# Genetic factors associated with reasons for clinical trial stoppage

Olesya Razuvayevskaya[1,2], Irene Lopez[1,2], Ian Dunham[1,2,3] & David Ochoa [1,2] ✉

Many drug discovery projects are started but few progress fully through clinical trials to approval. Previous work has shown that human genetics support for the therapeutic hypothesis increases the chance of trial progression. Here, we applied natural language processing to classify the free-text reasons for 28,561 clinical trials that stopped before their endpoints were met. We then evaluated these classes in light of the underlying evidence for the therapeutic hypothesis and target properties. We found that trials are more likely to stop because of a lack of efficacy in the absence of strong genetic evidence from human populations or genetically modified animal models. Furthermore, certain trials are more likely to stop for safety reasons if the drug target gene is highly constrained in human populations and if the gene is broadly expressed across tissues. These results support the growing use of human genetics to evaluate targets for drug discovery programs.

The drug discovery endeavor is dominated by high attrition rates, and failure remains the most likely outcome throughout the pipeline[1]. A diverse set of factors can lead to failure, with lack of efficacy or unforeseen safety issues reportedly explaining 79% of setbacks in the clinic[2]. New approaches adopted across the industry have aimed to improve success rates by systematically assessing the available evidence throughout the research and clinical pipelines[3,4]. Support from human genetic evidence has been repeatedly associated with successful clinical trial progression[5–8], ultimately supporting two-thirds of the drugs approved by the US Food and Drug Administration (FDA) in 2021 (ref. 9). Further understanding of the reasons for success or failure in clinical trials could assist in reducing future attrition.

Systematically assessing the reasons for success or failure in clinical trials can be hampered by many factors. Several surveys have demonstrated a bias towards reporting positive results, with 78.3% of trials in the literature reporting successful outcomes[10–12]. Successful clinical trials are published significantly faster than trials reporting negative results[13,14]. However, access to negative results is crucial, not only for revealing efficacy tendencies and safety liabilities[15] but also for retrospective review and benchmarking of predictive methods, including machine learning.

Since 2007, the FDA has required the submission of clinical trial results to ClinicalTrials.gov, a free-to-access global databank aimed at registering clinical research studies and their results[16,17]. For trials halted before their scheduled endpoint, ClinicalTrials.gov provides a freeform stopping reason: termination, suspension or withdrawal[18]. A team of researchers[19] previously classified the reasons for 3,125 stopped trials and found that only 10.8% of trials stopped because of a clear negative outcome. By contrast, the majority (54.5%) fell into a set of reasons characterized as neutral in relation to the therapeutic hypothesis, such as patient recruitment or other business or administrative reasons[19].

Here, we extended that work by training a natural language processing (NLP) model to classify stopping reasons and used this model to classify 28,561 stopped trials. We integrated our classification with evidence associating the drug target and disease from the Open Targets Platform[20], revealing that trials stopped for lack of efficacy or safety reasons were less supported by genetic evidence. Furthermore, oncology trials involving drugs for which the target gene is constrained in human populations were more likely to stop for safety reasons, whereas drugs with targets with tissue-selective expression were less likely to pose safety risks. These observations confirm and extend previous studies recognizing the value of genetic information and selective expression in target selection.

[1]Open Targets, Wellcome Genome Campus, Hinxton, Cambridgeshire, UK. [2]European Molecular Biology Laboratory, European Bioinformatics Institute (EMBL-EBI), Wellcome Genome Campus, Hinxton, Cambridgeshire, UK. [3]Wellcome Sanger Institute, Wellcome Genome Campus, Hinxton, Cambridgeshire, UK. ✉e-mail: ochoa@ebi.ac.uk

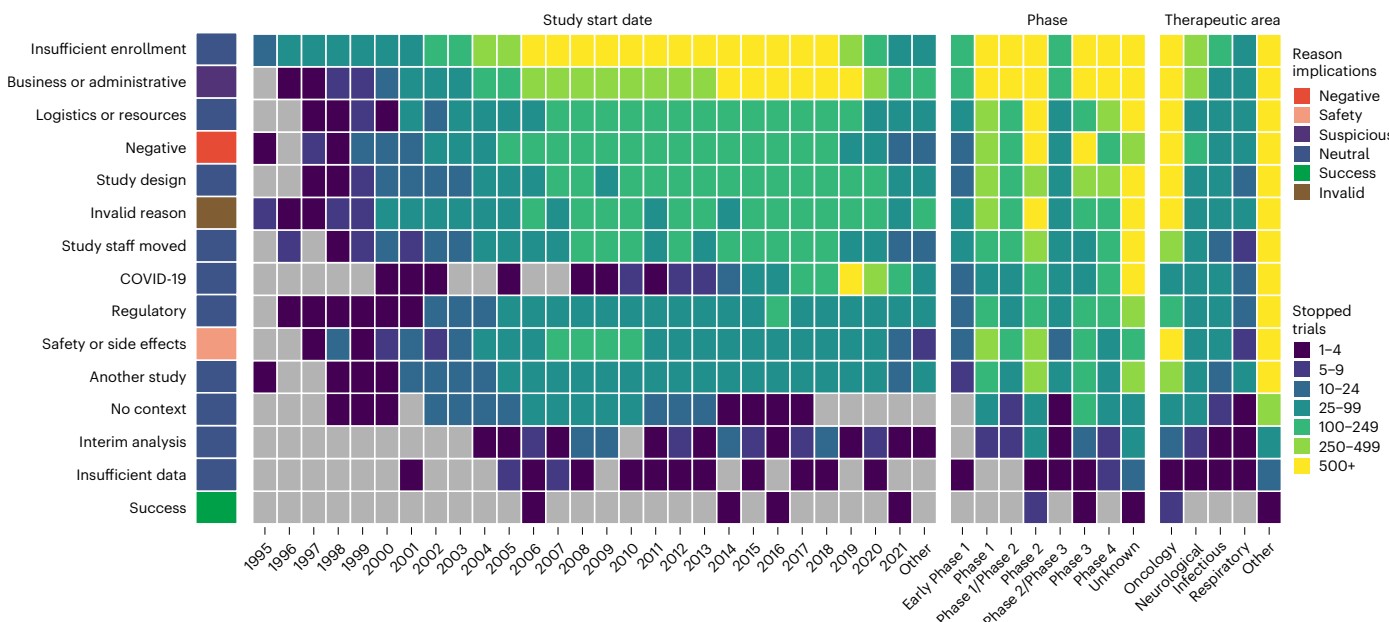

**Fig. 1 | Classification of stopped reasons for 28,561 clinical trials in ClinicalTrials.gov.** Predicted trial stop reasons are shown in rows with counts of trials per start year, clinical phase or therapeutic area shown by the color in each cell. The outcome groupings of the stopped reasons are shown using the color next to the stopped reason label. Note that trials start potentially many years before they are stopped.

## Results

### Interpretable classification of early stoppage reasons

To catalog the reasons behind the withdrawal, termination or suspension of clinical studies, we classified every free-text reason submitted to ClinicalTrials.gov using an NLP classifier. To build a training set for our model, we revisited the manual classification reported in a previous publication of 3,124 stopped trials based on the available submissions to ClinicalTrials.gov in May 2010 (ref. 19). The authors of that article classified every study with a maximum of three classes following an ontological structure (Supplementary Table 1). Each of the classes was also assigned a higher-level category representing the outcome implications for the clinical project. For example, 33.7% of the studies were classified as stopped owing to 'insufficient enrollment', a neutral outcome owing to its expected independence from the therapeutic hypothesis. When inspecting submitted reasons belonging to the same curated category, we observed a strong linguistic similarity, as revealed by clustering the cosine similarity of the sentence embeddings (Extended Data Fig. 1). Studies stopped because of reasons linked to lack of efficacy and studies stopped because of futility have a linguistic similarity of 0.98, with both classes manually classified as 'negative' outcomes. Based on this clustering, we redefined the classification by merging semantically similar classes represented by low numbers of annotated sentences. Moreover, we added 447 studies that were stopped as a result of the COVID-19 pandemic (Supplementary Table 2), resulting in a total of 3,571 studies manually classified into at least one of 17 stop reasons and explained by six different higher-level outcome categories.

By leveraging the consistent language used by the submitters, we fine-tuned the BERT model[21] for the task of clinical trial classification into stop reasons (Methods). Overall, the model showed strong predictive power in the cross-validated set ($F_{micro} = 0.91$), performing strongly for the most frequent classes, such as 'insufficient enrollment' ($F = 0.98$) or 'COVID-19' ($F = 1.00$), but demonstrating decreased performance on linguistically complex reasons, such as trials stopped because of another study ($F = 0.71$) (Supplementary Table 3).

To further evaluate the model, we manually curated an additional set of 1,675 stop reasons from randomly selected studies that were not included in the training set. Overall, the performance against the unseen data was lower but comparable to that of the cross-validated model ($F_{micro}$ ranging from 0.70 to 0.83 depending on the choice of the annotator) (Supplementary Table 4), demonstrating real-world performance and reduced risk of overfitting. Interestingly, the curators demonstrated a relatively low agreement for many classes in which the machine-learning model also showed relatively weak performance, such as studies stopped because of insufficient data or met endpoint (Methods and Extended Data Fig. 1).

### Reasons reflect operational, clinical and biological constraints

Classification of the 28,561 stopped trials submitted to ClinicalTrials.gov before 27 November 2021 was performed using our NLP model fine-tuned on all the manually curated sentences (Supplementary Table 5). In total, 99% of the trials were classified with at least one of the 15 potential reasons and mapped to one of six different higher-level outcomes (Fig. 1). 'Insufficient enrollment' remained the most common reason to stop a trial (36.67%), with other reasons before the accrual of any study results also occurring in a large number of studies. A total of 977 trials (3.38%) were classified as stopped because of 'safety or side effects', and 2,197 studies (7.6%) were stopped because of 'negative' reasons, such as those questioning the efficacy or value (futility). The incidence of each stop reason reflects the purpose of each phase (Extended Data Fig. 2). Studies stopped because of 'negative' outcomes more often impacted phase II (odds ratio (OR) = 1.9, $P = 2.4 \times 10^{-38}$) and phase III (OR = 2.6, $P = 3.64 \times 10^{-55}$), whereas studies stopped as a result of 'safety or side effects' declined in relative incidence after phase I (OR = 2.4, $P = 9.63 \times 10^{-23}$) (Supplementary Table 6). Trials stopped because of the relocation of the study or key staff occurred more than twice as often during early phase I, highlighting the importance of good clinical practices during the foundational stages. Of the studies that provided a stop reason, 48% were indicated for oncology. This large proportion is likely to be the combined result of the specific weight of oncology indications in the aggregated portfolio−27% of drug approvals in 2022−with the reported large incidence of clinical failures in oncology (32%) compared to other indications[22,23].

**a** Exposure: human genetic evidence     **b** Exposure: phenocopying mouse KO

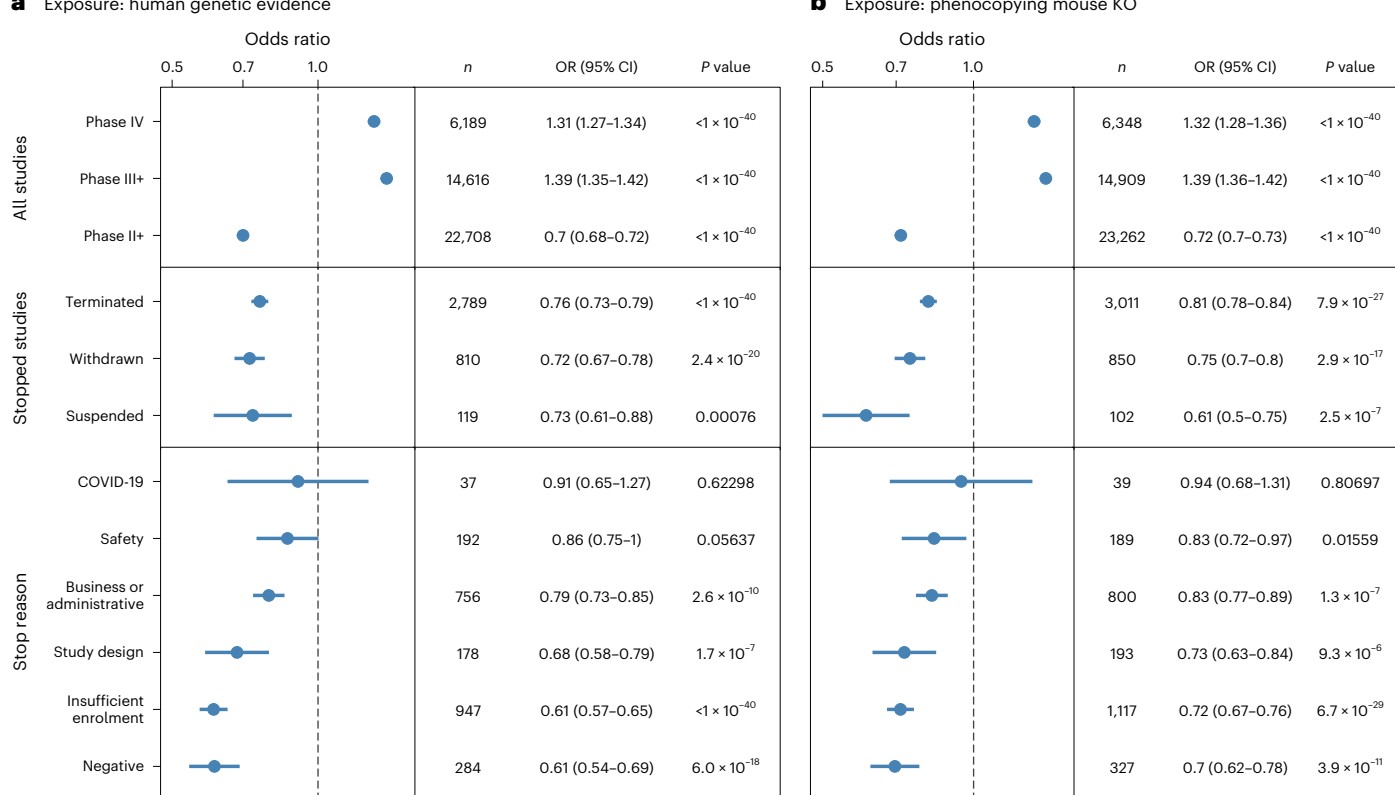

**Fig. 2 | Association between the availability of genetic evidence and clinical trial outcomes. a,b**, Genetic evidence support for clinical trials either from human genetics studies (**a**) or the International Mouse Phenotyping Consortium mouse knockouts (KO) that phenocopy the human disease (**b**). The panels show the odds ratio (OR) of support for the target-disease hypothesis from genetics evidence for all clinical trials split by phase (top row), stopped clinical trials (center row) and stopped clinical trials split by higher-level stopping reason (bottom row). The significance of the association between genetic evidence and trial outcome was assessed using a two-tailed Fisher's exact test, with a $P$ value threshold of 0.05 without multiple testing correction. The panels show the OR and 95% confidence intervals (CIs) for the association between genetic evidence and each subclass of trial. Significant ORs of >1 indicate enrichment and <1 indicate depletion.

Moreover, oncology studies stopped more frequently as a result of safety or side effects and were rarely stopped because of the COVID-19 pandemic (Extended Data Fig. 3). Alternatively, COVID-19 was the reported reason to stop respiratory studies at a higher rate than any other therapeutic area, possibly indicating increased operational difficulties.

**Genetic support for stopped trials influences the outcome**

To better understand the underlying reasons that might have caused the study to fail, we assessed the availability of different types of potentially causal genetic evidence for the intended pharmacological targets in the same indication (Extended Data Fig. 4). By using genetic evidence collated by the Open Targets Platform, we reproduced previous reports indicating that genetically supported studies are more likely to progress through the clinical pipeline (Fig. 2a)[5,6]. Interestingly, we also observed that stopped trials—among all the trials at any phase—are depleted in genetic support (OR = 0.73, $P = 3.4 \times 10^{-69}$). A similar lack of genetic evidence was observed for the three types of stopped studies: withdrawn, terminated and suspended (Supplementary Table 7).

When stratifying the stopped studies by reason, trials halted because of negative outcomes—such as lack of efficacy or futility—displayed a significant decrease of genetic support for the intended pharmacological target in the same indication (OR = 0.61, $P = 6 \times 10^{-18}$) (Fig. 2a). The depletion of genetic evidence on negative outcomes remains consistent when stratifying the indications by oncology (OR = 0.53) or non-oncology studies (OR = 0.75) (Extended Data Fig. 6), as well as when splitting by different sources of genetic evidence,

including genome-wide association studies processed by the Open Targets Genetics Portal[24], gene burden tests based on sequencing of large population cohorts[25–27], ClinVar[28], ClinGen Gene Validity[29], Genomics England PanelApp[30], gene2phenotype[31], Orphanet[32] and Uniprot[33] (Extended Data Fig. 5).

Other predicted reasons for stopping the trials, such as insufficient enrollment, problems with the study design or business or administrative reasons, also present a strong to moderate depletion of genetic evidence denoting potential reduced support for the therapeutic hypothesis (Fig. 2). We found that studies stopped as a result of coincidental factors such as the COVID-19 pandemic have no association with the availability of genetic support for the intended target in the primary indication.

The observed associations between clinical trial outcomes and the availability of genetic support remain consistent when considering genetic information in mouse models (Fig. 2b). Trials that were stopped because of negative factors present the weakest support among all predicted reasons (OR = 0.7, $P = 4 \times 10^{-11}$) when genetic evidence is defined as the presence of a murine model in which the drug target homologous gene knockout causes a phenotype that mimics the indication, as reported by the International Mouse Phenotyping Consortium[34].

**Genetic factors associated with safety-associated stopped trials**

Analysis of the classified stop reasons indicates that oncology trials are more likely to stop because of safety or side effects (OR = 2.14, $P = 8.1 \times 10^{-79}$; Supplementary Table 7). Moreover, for all trials predicted

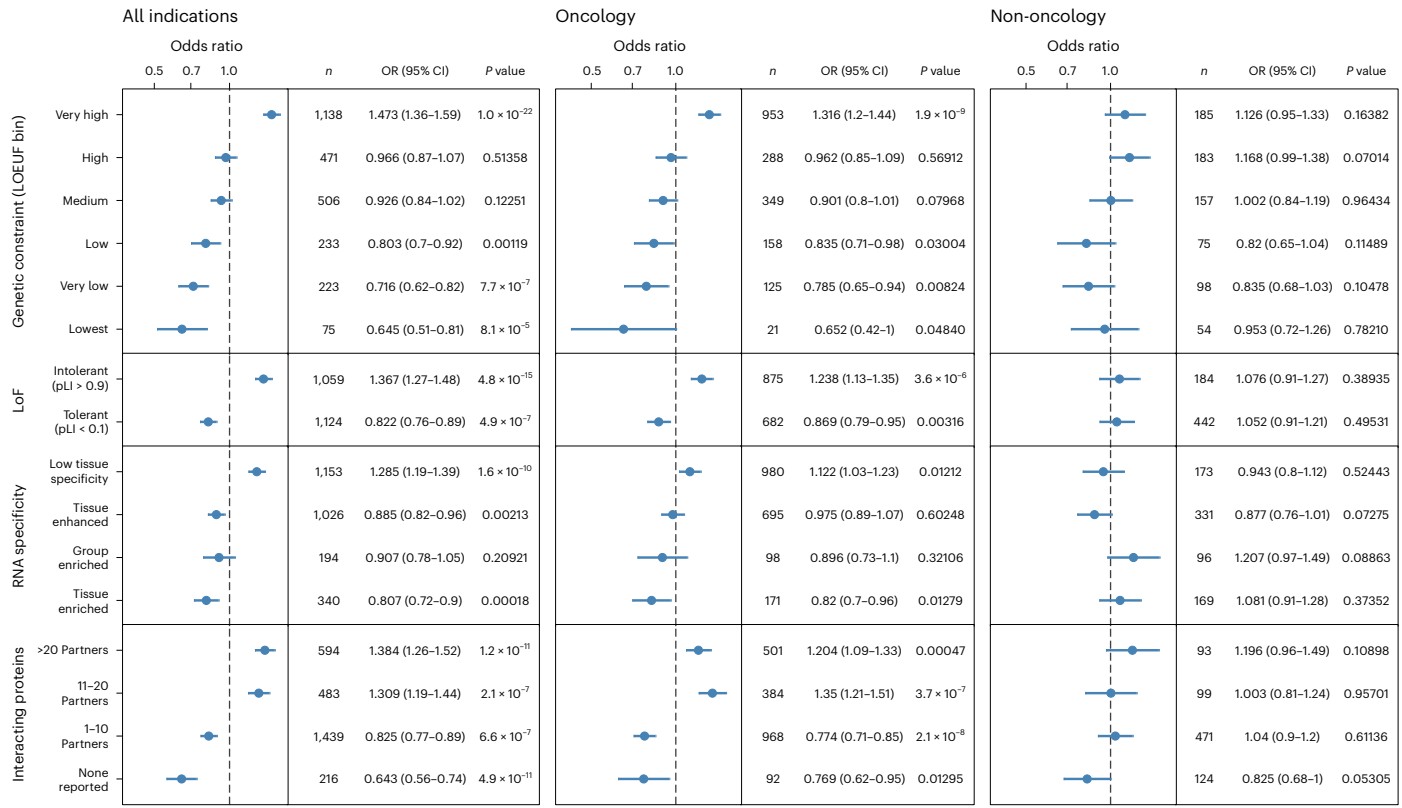

**Fig. 3 | Risk of stopping a clinical trial owing to safety or side effects examined by therapeutic area and target properties.** We evaluated the significance of the association between trials stopping because of safety or side effects and each variable (therapeutic area, relative genetic constraint as defined by GnomAD, tissue specificity as defined by the Human Protein Atlas and network connectivity with data from IntAct) with a two-tailed Fisher's exact test, with a P value threshold of 0.05 without multiple testing correction. OR > 1 represents an increased risk of study stopping and OR < 1 represents protection against stopping. Error bars, 95% CI; LoF, loss of function. Detailed results are presented in Supplementary Table 7.

to stop because of safety concerns, we found a significant enrichment in targets associated with driver events reported by COSMIC[35], ClinVar[28] or IntOgen[36] (Extended Data Fig. 7). Examining the target properties (Fig. 3), we found that studies targeting genes that are highly constrained in natural populations (GnomAD pLOEUF 16th percentile) are 1.5 times more likely to stop as a result of safety concerns[37]. Furthermore, the risk of stopping because of safety declines as the genetic constraint of the target decreases. Similarly, we identified a 1.4-fold increased risk of stopping because of safety concerns when the targeted gene is classified as loss-of-function intolerant (pLI > 0.9). These findings are compatible with previous evidence indicating that constrained genes are associated with increased side effects[38]. We also identified functional genomic features that inform on increased safety risk. According to the human protein atlas, a similar 1.3-fold increased risk is observed for genes expressed with low tissue specificity[39]. Instead, studies targeting tissue-enriched genes show a lower-than-expected (OR = 0.8, P = 1.8 × 10⁻⁴) likelihood of stopping because of safety. Finally, targets physically interacting with ten or more different partners according to the IntAct database (MI score > 0.42) present an increased risk of stopping as a result of safety concerns[40]. Further stratification of this analysis by indication denotes that these overall constraint signals impacting studies that are stopped because of safety are largely influenced by oncology trials.

## Discussion

Genetic evidence is increasingly leveraged by the pharmaceutical industry to add support to the therapeutic hypothesis[3,4,41,42]. Adding to previous observations on the role of genetic factors in overall trial success[5,6], we exploited under-used data from clinical trial records to better understand the opposite outcome: why clinical trials stop. Although the availability of genetic evidence might inform future success, failure remains the most common outcome of clinical studies, and, to our knowledge, no systematic evidence exists on the relevance of genetics to de-risk negative results.

Recent reports indicate that 79% of clinical studies fail because of a lack of efficacy or safety[2]. Our analysis indicates that within the 7.9% of studies that stop early because of withdrawal, termination or suspension, the proportion of trials that failed because of efficacy or safety is only 12.7%. Stopped studies are more likely to fail because of early coincidental factors that are not necessarily linked to biological plausibility; for example, the principal investigator relocates or there is insufficient enrollment in the trial. Notwithstanding the reduced relative risk of efficacy and safety as the main causes for stopping the trial, these studies provide a significant body of unsuccessful results that are probably explained by a weak therapeutic hypothesis. Continued expansion of the recording of negative results from clinical trials, including stoppages, will be valuable. To assist in this effort, we will continue to update the classification of stopped studies through the Open Targets Platform (https://platform.opentargets.org)[20]. Further investigation of the study outcomes for completed studies could expand our understanding of the reasons behind unsuccessful trials, particularly after accrual of the study results.

Our analysis exploits the classified stop reasons to understand the relative importance of the causes leading to failed studies. By using a case-control approach, we conclude that genetic support is not only predictive of clinical trial progression but also protective of early trial stoppage. We illustrate different ways in which genetic causality and genetic constraint can de-risk the target selection process.

However, many stopped trials, even when stopped for efficacy and safety reasons, might be explained by factors beyond the intended pharmacological target. Off-target effects, pharmacokinetics, drug delivery or toxicology are other risks not considered in this study that might also explain a set of negative outcomes. Another limitation of our study is that the reasons submitted to ClinicalTrials.gov might only represent a fraction of all the reasons contributing to the decision to halt the study. For example, we found that studies that were classified as stopped because of patient recruitment manifest weaker genetic support, an observation that we did not anticipate owing to the lack of an obvious link between enrollment and biological plausibility. Hence, we reason that a fraction of the stopped trials might present an overall lack of confidence in the therapeutic hypothesis, independently of the reported reason.

This study showcases how reflecting on past failures can inform the relative importance of the risks associated with early target identification and prioritization. Although clinical trial success is a discrete outcome, failure needs to be understood as a breakdown of many possible causes. A proper set of positive and negative outcomes such as the ones introduced in this work represent the groundwork necessary to implement quantitative or semi-automatic models to objectively de-risk any future studies.

## Online content

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

## Methods

### Inclusion and ethics

This study relied solely on aggregated genetic and clinical information available in public resources. It did not make use of individual-level data, and no specific ethics approval was required. Some of the data sources, including clinical study results, clinical curation of rare variants or genome-wide associations, might present biases towards European ancestries.

### NLP classification of stopped clinical trials

To quantify the semantic structure of the reasons for clinical trial stop, we analyzed the classification of the stopped clinical trials developed in a previous publication[19]. We trained a long short-term memory network to create the representations for each stop reason and averaged the embeddings across all examples of a particular class. The class embeddings were then used to calculate the cosine similarities among classes and were visualized using agglomerative hierarchical clustering (Extended Data Fig. 1). The hierarchical representation illustrates the clusters that are semantically close to each other, along with the number of examples per class and the parent category. Similar classes with fewer sentences were grouped together to ensure representative categories based on clinical expertise and semantic similarity. For example, the classes 'study moved' and 'key staff left' are semantically clustered together and attend to similar underlying reasons. The list of categories defined in the previous publication and their redefined groupings can be found in Supplementary Table 8.

To validate the model on new data and expand the training set, we performed a human annotation experiment of 1,675 additional ClinicalTrials.gov studies that were not classified in the previous publication. We randomly assigned six sets of 250 unique stopped trials to each curator, including 25 overlapping trials, to estimate the inter-annotator agreement. Across 3 pairs of annotators, we estimated inter-annotator agreements of 0.8, 0.71 and 0.66 using the kappa statistic[43].

### Stop reason classification model

We fine-tuned the BERT model for the task of predicting the stop reasons on the training set of 4,500 human-annotated stopped clinical trials[21]. We used a BERT uncased pre-trained model with a one-layer feed-forward classifier consisting of a ReLU layer between the input and output layers, in which the input and output layers represent linear layers. Fine-tuning was performed by using the HuggingFace transformer library[44]. The classifier uses 50 hidden units and the ReLU activation function.

We used the last hidden state at token '[CLS]' to retrieve a representation of the whole explanation and fed it into the classifier. We then applied 'sigmoid' over the logits to retrieve the probabilities. The best accuracy on the validation set was achieved while training the model for seven epochs with a batch size of 32, a learning rate of $5 \times 10^{-5}$ and the Pytorch implementation of Adam's optimizer with weights decay, in which the weight decay is set to the default value of $1 \times 10^{-2}$. The test set was created stratified to ensure that the relative class frequencies were considered in each fold of the test set. Given that the nature of the task does not assume that the categories are mutually exclusive and the original and new annotation tasks allowed human annotators to mark up to three categories, we treated the top three probabilities returned by the model that are above a pre-defined threshold as correct answers.

### Clinical studies

We collated all clinical trials from ClinicalTrials.gov as of 27 November 2021 and classified the 28,561 stopped studies (withdrawn, suspended or terminated). Genetic traits and indications from clinical studies were harmonized using the Experimental Factor Ontology (EFO)[45]. When studies contained multiple indications, their similarity based on the EFO structure was evaluated. All indications were considered when indications were similar (for example, several oncology indications).

When indications were dissimilar (for example, diabetes in malaria patients), the diseases were curated to annotate the appropriate indication for the study. Drugs reported as approved by the FDA were also considered to ensure the representation of medicines preceding the ClinicalTrials.gov resource. To map each drug or clinical candidate to its pharmacological targets, we leveraged the molecule mechanism of action from the ChEMBL database[46]. All possible annotations were used if a drug could be mapped to multiple targets. All drug targets were annotated against Ensembl gene IDs[47] when possible. To perform subsequent analyses, only drugs with a known mechanism of action were considered. The resulting dataset contains 594,375 clinical target-disease records, capturing 71,419 unique target-disease associations and 57,775 target-disease pairs in studies that stopped early[48].

### Target-disease genetic support

We integrated 13 sources available in the Open Targets Platform in April 2022 to extract a comprehensive list of genetically supported gene–disease associations. The genetic evidence was mapped to Ensembl gene identifiers and EFO identifiers as part of the Open Targets activities. To represent common disease genetics, we leveraged Open Targets Genetics[24], a post-genome-wide association study analysis leveraging different functional genomics features. In this study, we used all gene assignments based on a locus-to-gene score above 0.05 (ref. 49). The other predominantly germline genetic sources included in this analysis are Gene Burden[25–27], ClinVar[28], Genomics England PanelApp[30], Gene2Phenotype[31], Clingen Gene–Disease Validity[29], Uniprot[33] and Orphanet[32]. We included COSMIC Cancer Hallmarks[35], IntOgen cancer drivers[36] and ClinVar somatic variants[50] as sources of somatic genetic evidence. As a source to capture the effects of genetic variation in animal models, we included the mouse–human phenotypic mappings reported by the International Mouse Phenotyping Consortium[34]. Genetic evidence was ontologically expanded using the EFO, resulting in 3,654,109 genetically supported gene–trait pairs. This dataset represents a redundant view of the evidence, with its only purpose being to maximize the overlap with the clinical information and minimize the issues related to the sparsity in the annotation.

### Target annotations

To analyze the target factors that could influence studies stopped because of safety or side effects, we also included a set of target annotations that were independent of the study indication. Each gene was annotated with genetic constraint data from gnomAD, representing the functional impact of the presence of genetic variants, and split into six categories derived from gnomAD's pLOEUF quantiles. We also analyzed the predicted loss-of-function intolerance, distinguishing genes as 'LoF-intolerant' when the pLI score is above 0.9 and as 'LoF tolerant' when the pLI score was below 0.1 (ref. 37). Moreover, each target was classified in a bin based on the number of unique interacting partners above an MI score threshold of 0.42 in the IntAct database[40]. This threshold corresponds to a physical interaction identified at least once in low-throughput studies or replicated in multiple high-throughput experiments. Additionally, target annotation for tissue specificity and distribution was retrieved from the baseline transcriptomic experiments in the Human Protein Atlas database[39]. The assessment was performed according to the categories defined by the Human Protein Atlas.

### Statistics and reproducibility

No preliminary statistical analyses were conducted to determine sample sizes. The choice of clinical studies and genetic information followed an unbiased procedure. The significance of each case-control study was computed using a two-sided Fisher's exact test using all available samples. No multiple testing correction was applied to the resulting $P$ values. All statistical tests were computed using SciPy (v.1.11.4)[50]. The code to replicate the analyses is publicly available (see Code availability).

**Reporting summary**

Further information on research design is available in the Nature Portfolio Reporting Summary linked to this article.

## Data availability

The full training set is available for download at HuggingFace, including the curation from the previously published article[19] and the COVID-19 stopped studies[51]. The resulting model for download or interactive exploration can also be found in HuggingFace[52]. The dataset of the clinical trial stop reason predictions used in this study is available in Github[53]. The collection of clinical studies annotated with predicted stop reasons and genetic evidence can also be accessed on Hugging-Face. Up-to-date predictions for newer clinical trial studies are updated quarterly in the Open Targets Platform.

## Code availability

Code to reproduce the model and analysis are available on GitHub[53].

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

## Acknowledgements

We would like to thank T. R. Pak, M. D. Rodriguez and F. P. Roth from Harvard Medical School, the Dana-Farber Cancer Institute and the Donelly Center (University of Toronto) for providing the dataset of curated stop reasons that was used for training our model. We also thank the Open Targets team for manually curating the stop reasons for the additional set of 1,675 clinical studies, including A. Hercules, A. Gonzalez, K. Tsirigos and H. Cornu. Finally, we would like to thank S. Machlitt-Northen from GlaxoSmithKline for providing detailed feedback on the redefined categories for stopped trials. I.D.'s research was funded in part by a Wellcome Trust grant (grant number 206194). For the purpose of Open Access, the authors have applied a CC-BY public copyright license to any author-accepted manuscript version arising from this submission.

## Author contributions

O.R, I.D. and D.O. designed the study. O.R. and I.L. trained the models. O.R., I.L. and D.O. conducted the analysis. O.R., I.L., I.D. and D.O. wrote the manuscript.

## Funding

## Competing interests

The authors declare no competing interests.

## Additional information

**Extended data** is available for this paper at https://doi.org/10.1038/s41588-024-01854-z.

**Correspondence and requests for materials** should be addressed to David Ochoa.

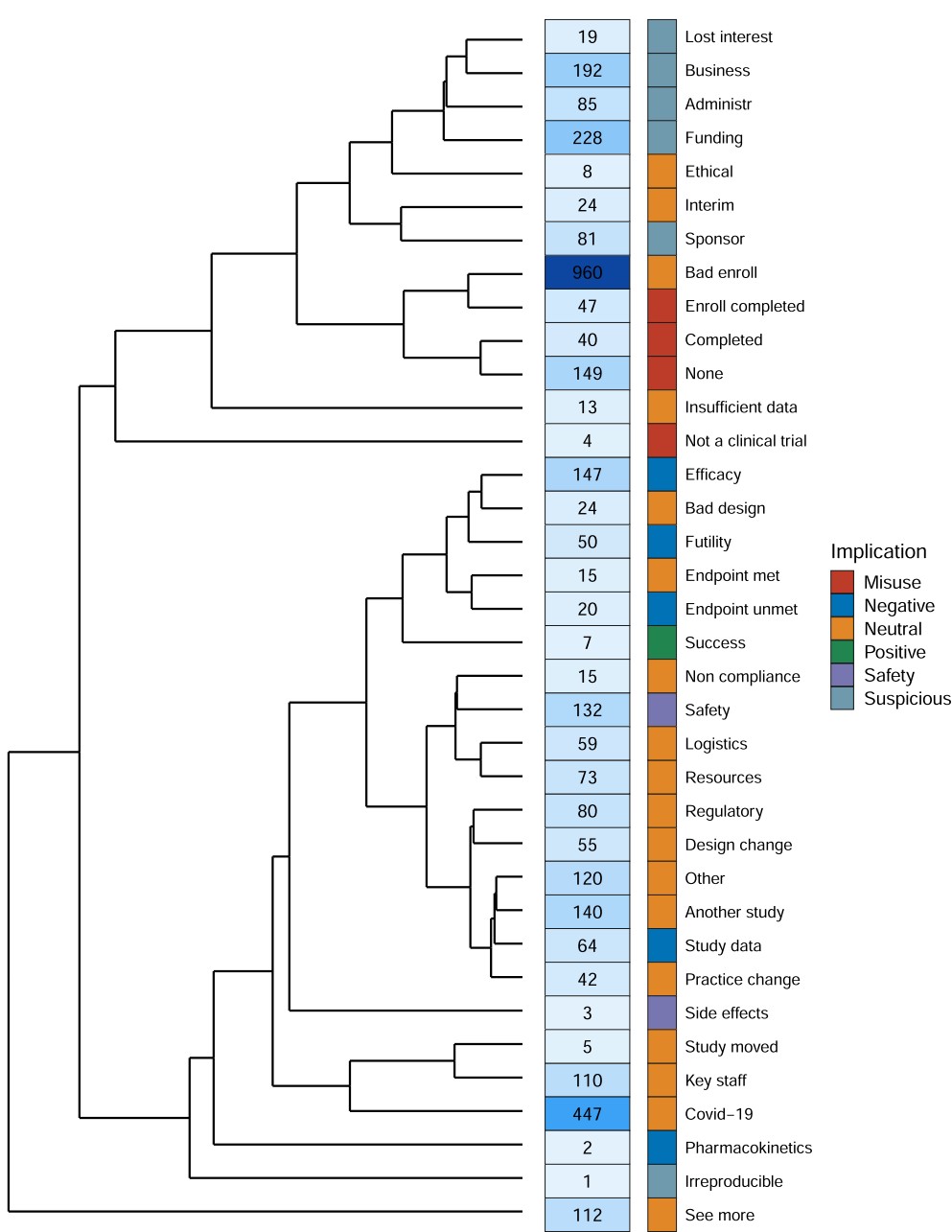

**Extended Data Fig. 1 | Hierarchical clustering of stop reason similarity based on curation from Pak et al. and 447 additional stopped trials due to COVID-19.**
Distances were estimated as the cosine similarity of the averaged embeddings (see Methods section).

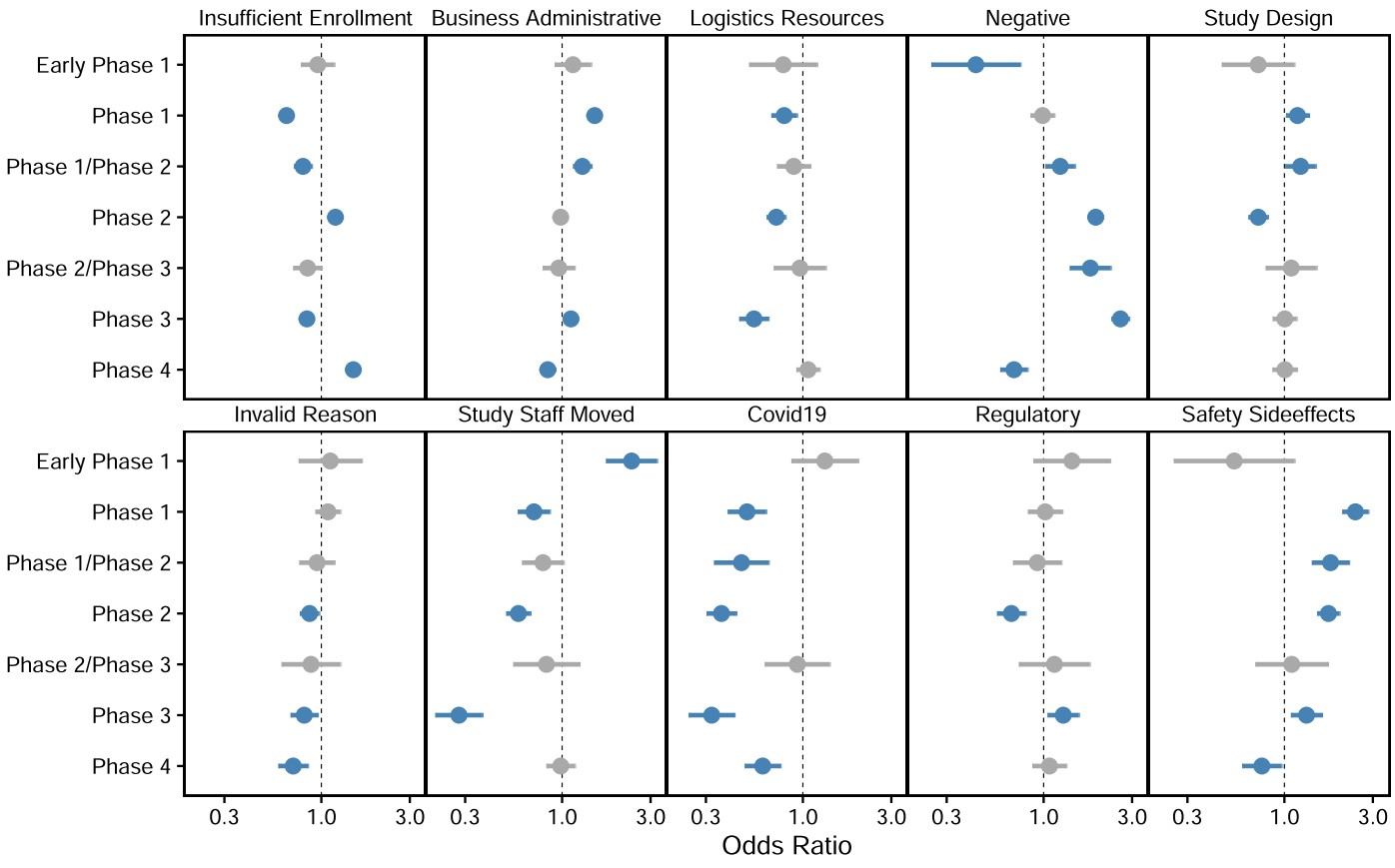

**Extended Data Fig. 2 | Association between the reasons to stop along each phase of the clinical development as reported by ClinicalTrials.gov.** Underpowered reasons for stoppage were excluded. We used a two-tailed Fisher's exact test to assess the significance of the associations, with a p-value threshold of 0.05 (n = 10,214 independent stopped trials). Significant associations are highlighted in blue. Error bars represent 95% confidence intervals for the odds ratio. All results are provided in Supplementary Table 6.

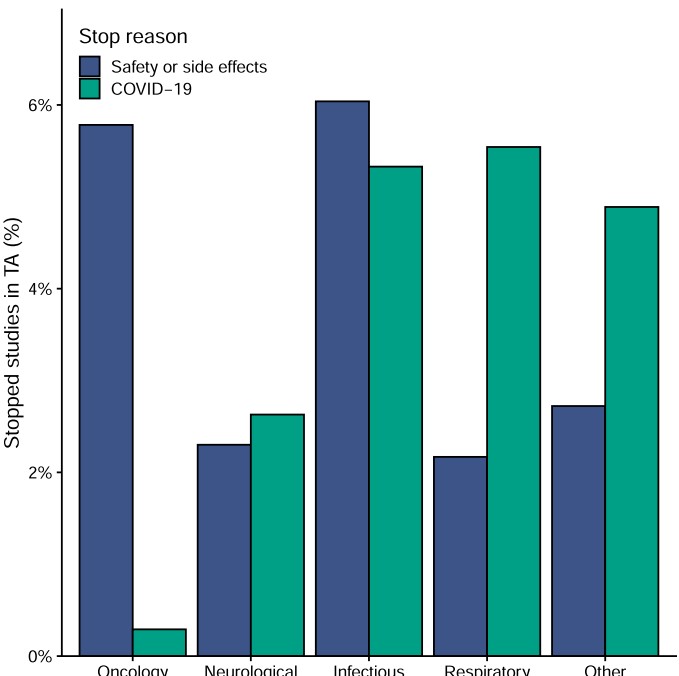

**Extended Data Fig. 3 | Percentage of stopped trials predicted to be halted due to Safety or side effects or the COVID-19 pandemic as a fraction of all the trials by predominant therapeutic area.** Indications with multiple possible therapeutic areas were associated with the most severe area (for example Oncology). Overall incidence when considering all therapeutic areas was 5% for COVID-19 and 3.3% for Safety or side effects.

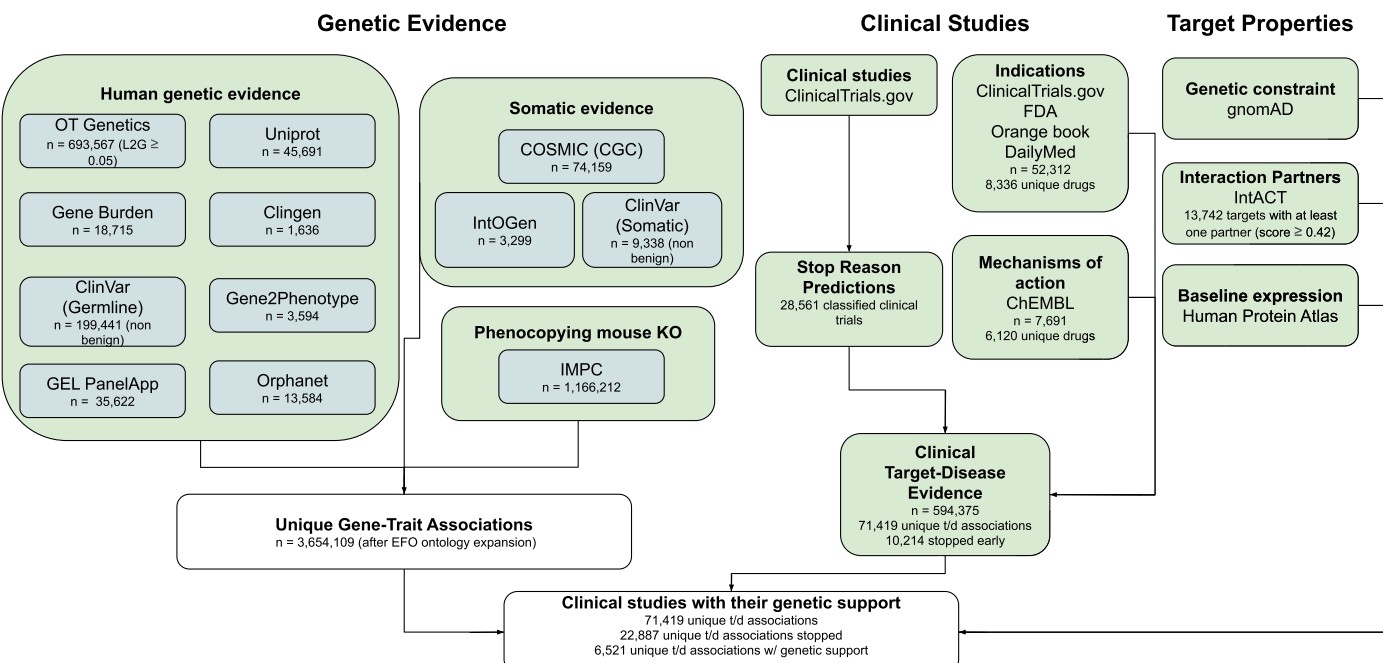

**Extended Data Fig. 4 | Representation of the data and analytical workflow defined to investigate the predictive value of genetics in all target/disease associations derived from clinical trials.** Except for the baseline expression data, all datasets were sourced from the Open Targets 22.04 release. Detailed methods can be found in the Methods section.

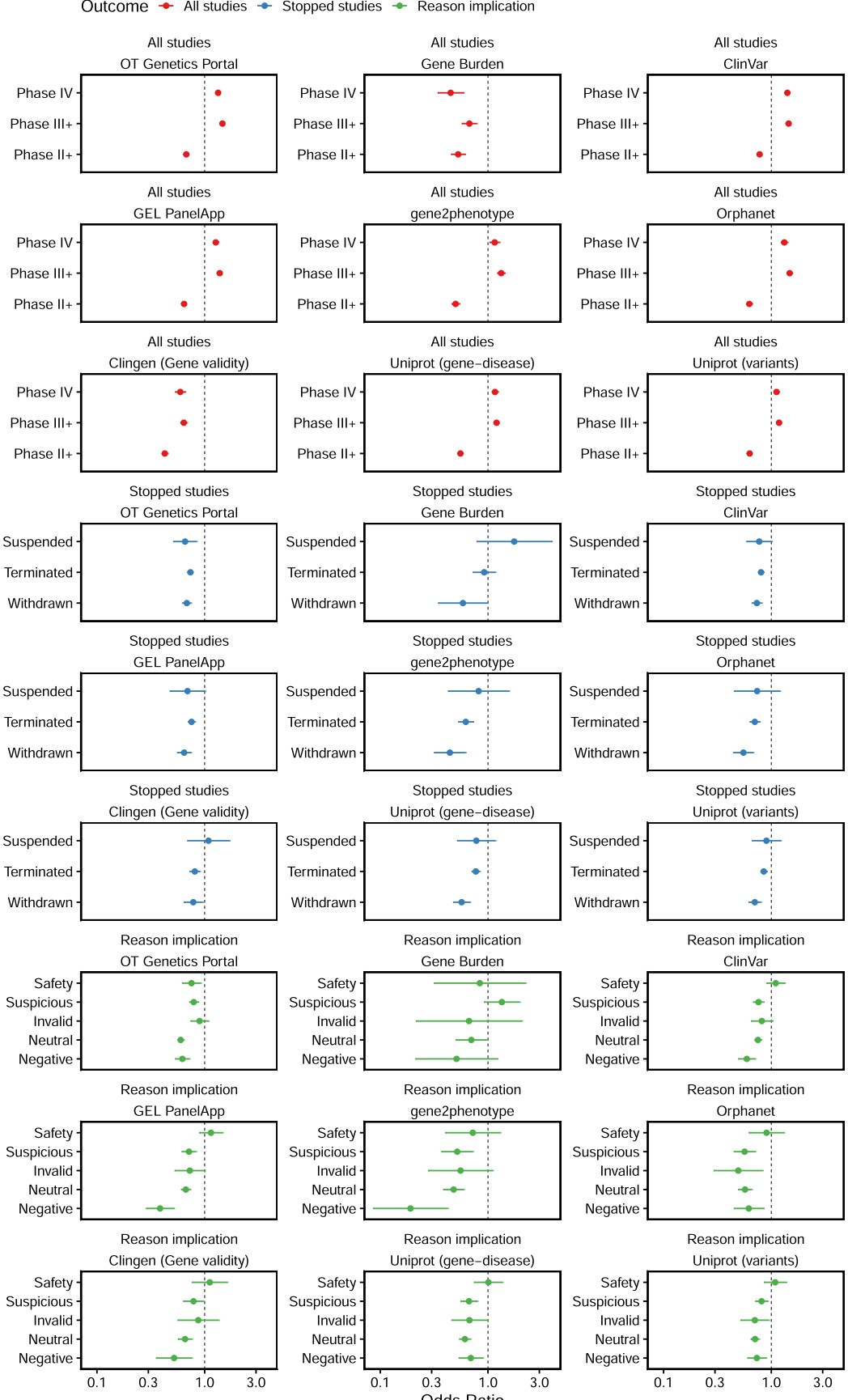

**Extended Data Fig. 5 | Association between the availability of genetic evidence and clinical trial outcomes by genetic data source.** X-axis displays the respective odds ratio and y-axis groups the studies by phase (red), stopped clinical trials (blue) and stopped clinical trials split by high-level stopping reason (green). Error bars represent 95% confidence intervals for the odds ratio.

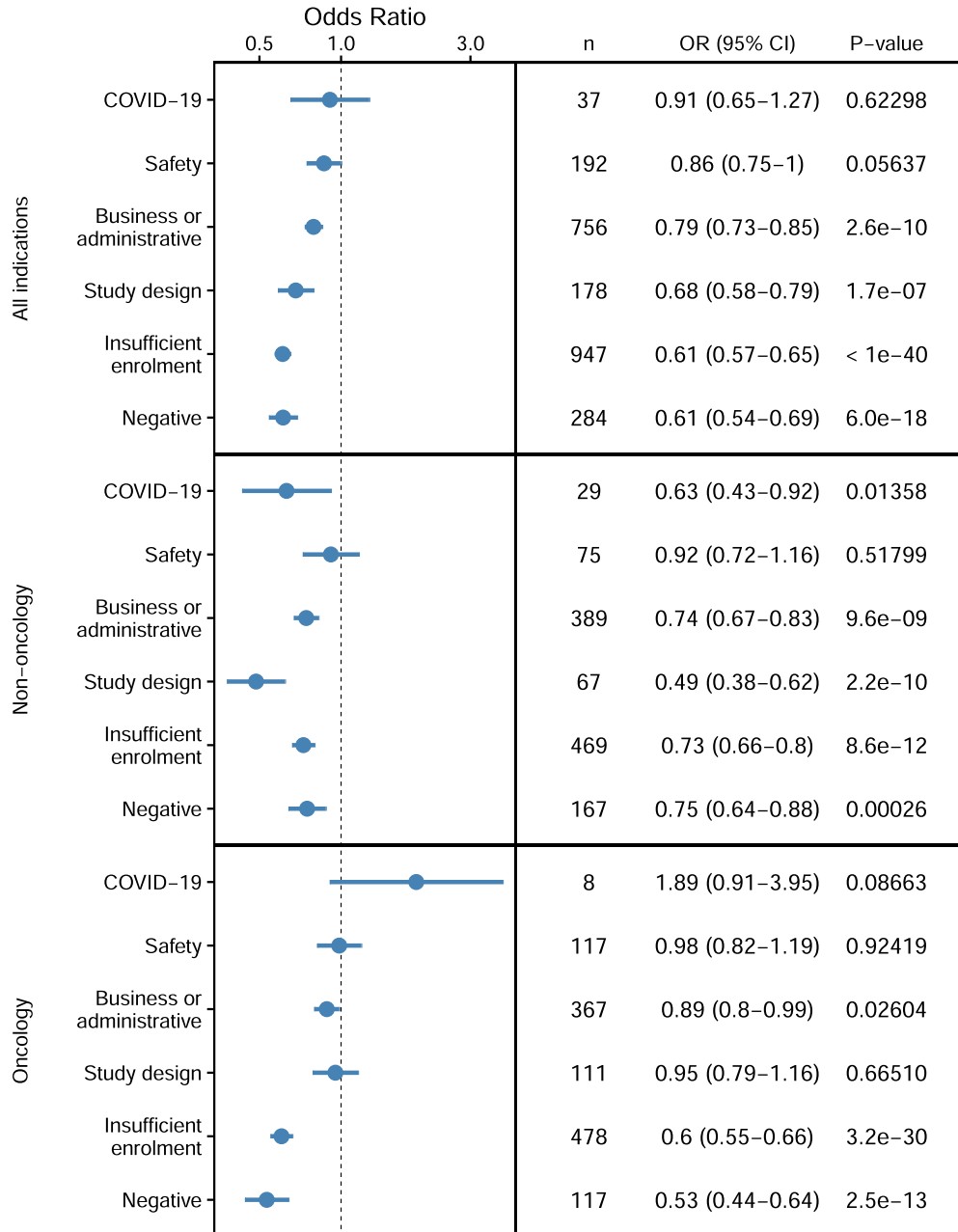

| | Odds Ratio | n | OR (95% CI) | P-value |
|---|---|---|---|---|
| **All indications** | | | | |
| COVID-19 | | 37 | 0.91 (0.65–1.27) | 0.62298 |
| Safety | | 192 | 0.86 (0.75–1) | 0.05637 |
| Business or administrative | | 756 | 0.79 (0.73–0.85) | 2.6e–10 |
| Study design | | 178 | 0.68 (0.58–0.79) | 1.7e–07 |
| Insufficient enrolment | | 947 | 0.61 (0.57–0.65) | < 1e–40 |
| Negative | | 284 | 0.61 (0.54–0.69) | 6.0e–18 |
| **Non-oncology** | | | | |
| COVID-19 | | 29 | 0.63 (0.43–0.92) | 0.01358 |
| Safety | | 75 | 0.92 (0.72–1.16) | 0.51799 |
| Business or administrative | | 389 | 0.74 (0.67–0.83) | 9.6e–09 |
| Study design | | 67 | 0.49 (0.38–0.62) | 2.2e–10 |
| Insufficient enrolment | | 469 | 0.73 (0.66–0.8) | 8.6e–12 |
| Negative | | 167 | 0.75 (0.64–0.88) | 0.00026 |
| **Oncology** | | | | |
| COVID-19 | | 8 | 1.89 (0.91–3.95) | 0.08663 |
| Safety | | 117 | 0.98 (0.82–1.19) | 0.92419 |
| Business or administrative | | 367 | 0.89 (0.8–0.99) | 0.02604 |
| Study design | | 111 | 0.95 (0.79–1.16) | 0.66510 |
| Insufficient enrolment | | 478 | 0.6 (0.55–0.66) | 3.2e–30 |
| Negative | | 117 | 0.53 (0.44–0.64) | 2.5e–13 |

**Extended Data Fig. 6 | Genetic support for stopped clinical trials in all indications, non-oncology and oncology.** Each row represents a stopping reason, with the effect size in the form of odds ratio (OR) and its 95% confidence interval represented by the dot and error bar. An odds ratio (OR) > 1 suggests that trials stopped for a given reason are more likely to have genetic support, while an OR < 1 indicates depletion. The number of trials (n) supporting each estimate is provided. Statistical significance was assessed using a two-tailed Fisher's exact test, with a significance threshold of p < 0.05 without multiple testing correction.

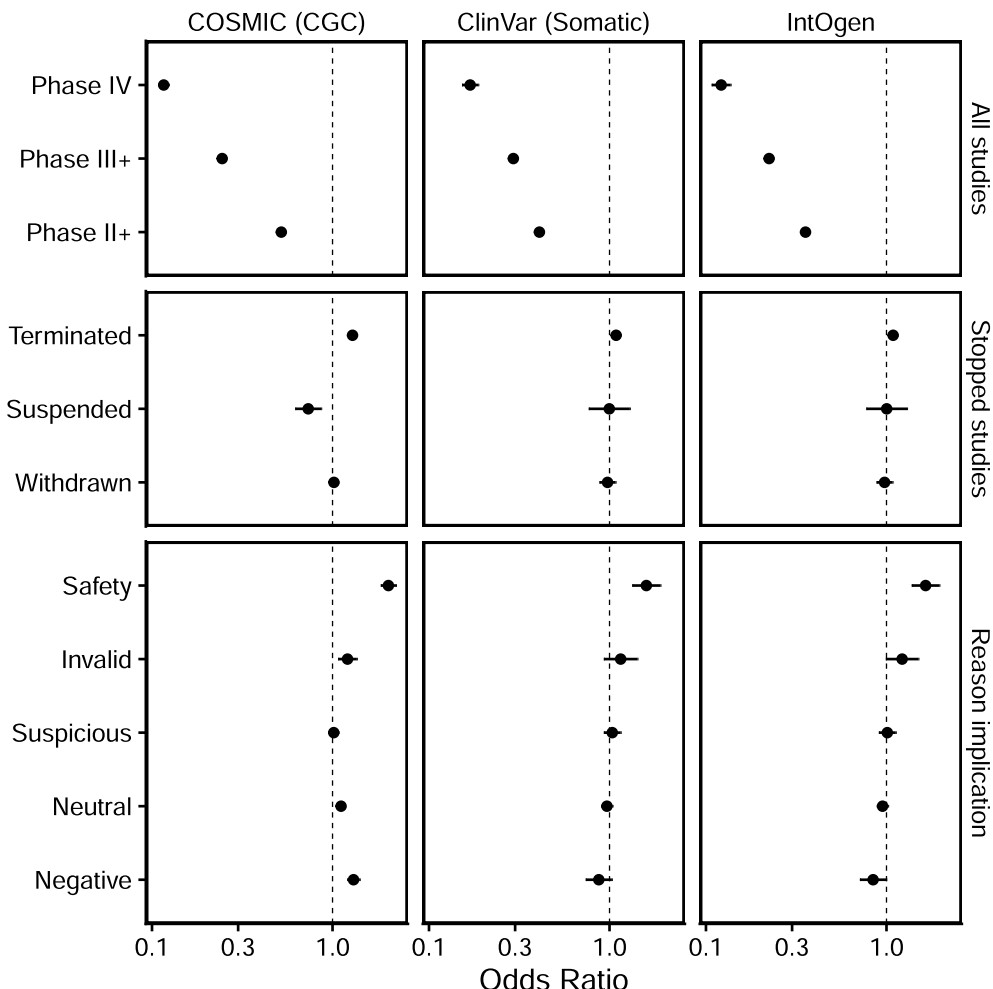

**Extended Data Fig. 7 | Association between the availability of genetic evidence and clinical trial outcomes by somatic data source.** X-axis displays the respective odds ratio and y-axis groups the studies by phase (top row), stopped clinical trials (centre row) and stopped clinical trials split by high-level stopping reason (bottom row). Error bars represent 95% confidence intervals for the odds ratio.

# Reporting Summary

## Statistics

For all statistical analyses, confirm that the following items are present in the figure legend, table legend, main text, or Methods section.

| n/a | Confirmed | |
|---|---|---|
| ☐ | ☒ | The exact sample size (*n*) for each experimental group/condition, given as a discrete number and unit of measurement |
| ☒ | ☐ | A statement on whether measurements were taken from distinct samples or whether the same sample was measured repeatedly |
| ☐ | ☒ | The statistical test(s) used AND whether they are one- or two-sided<br>*Only common tests should be described solely by name; describe more complex techniques in the Methods section.* |
| ☒ | ☐ | A description of all covariates tested |
| ☐ | ☒ | A description of any assumptions or corrections, such as tests of normality and adjustment for multiple comparisons |
| ☐ | ☒ | A full description of the statistical parameters including central tendency (e.g. means) or other basic estimates (e.g. regression coefficient) AND variation (e.g. standard deviation) or associated estimates of uncertainty (e.g. confidence intervals) |
| ☐ | ☒ | For null hypothesis testing, the test statistic (e.g. *F*, *t*, *r*) with confidence intervals, effect sizes, degrees of freedom and *P* value noted<br>*Give P values as exact values whenever suitable.* |
| ☒ | ☐ | For Bayesian analysis, information on the choice of priors and Markov chain Monte Carlo settings |
| ☒ | ☐ | For hierarchical and complex designs, identification of the appropriate level for tests and full reporting of outcomes |
| ☒ | ☐ | Estimates of effect sizes (e.g. Cohen's *d*, Pearson's *r*), indicating how they were calculated |

*Our web collection on statistics for biologists contains articles on many of the points above.*

## Software and code

Policy information about availability of computer code

| Data collection | All the code necessary to prepare the data for analysis is available in Github doi:10.5281/ZENODO.11966097. |
|---|---|
| Data analysis | All the code necessary to perform the analysis is available in Github doi:10.5281/ZENODO.11966097. |

For manuscripts utilizing custom algorithms or software that are central to the research but not yet described in published literature, software must be made available to editors and reviewers. We strongly encourage code deposition in a community repository (e.g. GitHub). See the Nature Portfolio guidelines for submitting code & software for further information.

## Data

Policy information about availability of data

All manuscripts must include a data availability statement. This statement should provide the following information, where applicable:
- Accession codes, unique identifiers, or web links for publicly available datasets
- A description of any restrictions on data availability
- For clinical datasets or third party data, please ensure that the statement adheres to our policy

The full training set is available for download at HuggingFace, including the Pak et al. curation and the COVID-19 stopped studies (10.57967/HF/2600). The resulting model for download or interactive exploration can also be found in HuggingFace (10.57967/HF/2599 ). Freeze of the clinical trial stop reason predictions used in this study is available in Github (10.5281/ZENODO.11966097). The collection of clinical studies annotated with predicted stop reasons and genetic evidence can also be accessed on Hugging Face. Up-to-date predictions for newer clinical trial studies are updated quarterly in the Open Targets Platform.

## Research involving human participants, their data, or biological material

Policy information about studies with [human participants or human data](). See also policy information about [sex, gender (identity/presentation), and sexual orientation]() and [race, ethnicity and racism]().

| | |
|---|---|
| Reporting on sex and gender | N/A |
| Reporting on race, ethnicity, or other socially relevant groupings | N/A |
| Population characteristics | N/A |
| Recruitment | N/A |
| Ethics oversight | N/A |

Note that full information on the approval of the study protocol must also be provided in the manuscript.

# Field-specific reporting

Please select the one below that is the best fit for your research. If you are not sure, read the appropriate sections before making your selection.

☒ Life sciences  ☐ Behavioural & social sciences  ☐ Ecological, evolutionary & environmental sciences

For a reference copy of the document with all sections, see [nature.com/documents/nr-reporting-summary-flat.pdf](nature.com/documents/nr-reporting-summary-flat.pdf)

# Life sciences study design

All studies must disclose on these points even when the disclosure is negative.

| | |
|---|---|
| Sample size | Based on the availability of clinical studies for each comparison. More details in each individual test. |
| Data exclusions | No data exclusions |
| Replication | Not applicable |
| Randomization | Not applicable |
| Blinding | Not applicable |

# Reporting for specific materials, systems and methods

We require information from authors about some types of materials, experimental systems and methods used in many studies. Here, indicate whether each material, system or method listed is relevant to your study. If you are not sure if a list item applies to your research, read the appropriate section before selecting a response.

### Materials & experimental systems

| n/a | Involved in the study |
|---|---|
| ☒ ☐ | Antibodies |
| ☒ ☐ | Eukaryotic cell lines |
| ☒ ☐ | Palaeontology and archaeology |
| ☒ ☐ | Animals and other organisms |
| ☒ ☐ | Clinical data |
| ☒ ☐ | Dual use research of concern |
| ☒ ☐ | Plants |

### Methods

| n/a | Involved in the study |
|---|---|
| ☒ ☐ | ChIP-seq |
| ☒ ☐ | Flow cytometry |
| ☒ ☐ | MRI-based neuroimaging |

## Plants

Seed stocks          N/A

Novel plant genotypes          N/A

Authentication          N/A

