## [Peer Review File · Nature Genetics]

Peer Review Information

Manuscript Title: Genetic factors associated with reasons for clinical trial stoppage

Corresponding author name(s): Dr David Ochoa

Reviewer Comments & Decisions:

Decision Letter, revise:

1st Mar 2023

Dear Dr Ochoa,

Thank you for submitting your manuscript entitled "Why Clinical Trials Stop: The Role of Genetics".

As previously communicated, we have given the paper our careful consideration and find it of potential interest. However, we are unable to send your study out for peer review without the Editorial Policy Checklist and Reporting Summary (see links below).

We shall hope to receive your revised version as soon as you are able to complete the checklists. If something similar is published in the interim we will have to consider the impact it has on the novelty of a revised manuscript.

If you anticipate a delay of more than four weeks, please let us know. We will be happy to consider your revision so long as nothing similar has been accepted for publication at Nature Genetics or published elsewhere. Should your manuscript be substantially delayed without notifying us in advance and your article is eventually published, the received date may be that of the revised, not the original, version.

If you are not interested in submitting a suitably revised manuscript in the future please let me know immediately so we can close your file. If you have any questions, please contact me.

1) Please ensure that you have completed the Reporting Summary required for review:
<https://www.nature.com/documents/nr-reporting-summary.pdf>

2) Please also complete the Editorial Policy Checklist that would be a requirement for eventual publication in a Nature journal:
<https://www.nature.com/documents/nr-editorial-policy-checklist.pdf>

Please be aware of our guidelines on digital image standards.

Please use the link below when you are prepared to resubmit.
[redacted]

Thank you for your interest in Nature Genetics.

Sincerely,

Michael Fletcher, PhD
Senior Editor, Nature Genetics

ORCID: 0000-0003-1589-7087

Decision Letter, initial version:

31st Mar 2023

Dear Dr Ochoa,

Your Article, "Why Clinical Trials Stop: The Role of Genetics" has now been seen by 3 referees. You will see from their comments below that while they find your work of interest, some important points are raised. We are interested in the possibility of publishing your study in Nature Genetics, but would like to consider your response to these concerns in the form of a revised manuscript before we make a final decision on publication.

In brief, the three reviews all acknowledge the potential broad interest in your work, but are split on the overall robustness of your analysis.

Reviewer #1 highlights a major issue: they suggest that the large number of oncology trials is skewing the results of your analysis and that this must be thoroughly investigated before your analysis can be reviewed in its entirety.

We note that Referee #2 also comments on the preponderance of oncology trials - suggesting this is an especially important issue to address in a revision - but otherwise sounds quite positive.

Reviewer #3 is also, in the main, positive and has also makes a few suggestions for new analysis.

To guide the scope of the revisions, the editors discuss the referee reports in detail within the team, including with the chief editor, with a view to identifying key priorities that should be addressed in revision and sometimes overruling referee requests that are deemed beyond the scope of the current study. We hope that you will find the prioritized set of referee points to be useful when revising your study. Please do not hesitate to get in touch if you would like to discuss these issues further.

We therefore invite you to revise your manuscript taking into account all reviewer and editor comments. Please highlight all changes in the manuscript text file. At this stage we will need you to upload a copy of the manuscript in MS Word .docx or similar editable format.

*2) If you have not done so already please begin to revise your manuscript so that it conforms to our Article format instructions, available here.

*3) Include a revised version of any required Reporting Summary:

Please be aware of our guidelines on digital image standards.

[redacted]

Nature Genetics is committed to improving transparency in authorship. As part of our efforts in this

direction, we are now requesting that all authors identified as 'corresponding author' on published papers create and link their Open Researcher and Contributor Identifier (ORCID) with their account on the Manuscript Tracking System (MTS), prior to acceptance. ORCID helps the scientific community achieve unambiguous attribution of all scholarly contributions. You can create and link your ORCID from the home page of the MTS by clicking on 'Modify my Springer Nature account'. For more information please visit www.springernature.com/orcid.

Sincerely,

Michael Fletcher, PhD
Senior Editor, Nature Genetics

ORCID: 0000-0003-1589-7087

Referee expertise:

Referee #1: genetics; clinical trials; genomics.

Referee #2: genetics/genomics; clinical trials.

Referee #3: genetics, genomics, machine learning.

Reviewers' Comments:

Reviewer #1:

Remarks to the Author:

Razuvayevskaya et al. present the training of a natural language processing model to identify the reason for stopping clinical trials in the large Clinicaltrials.gov database and analysis of some of the genetic and genomic factors that influence those stopping reasons. From the validation exercise carried out by the authors, they have created a valuable model and classified data set that will support other research in this area. Their analyses of this data set may provide valuable insights that can impact drug discovery and development decisions.

Based on the modest amount of methodological description in the paper, I am concerned with the robustness of the conclusions the authors draw from their analyses. My greatest concerns are 1) ignoring the impact that such a large fraction of oncology studies may have on the primary study measures and 2) the lack of positive trial outcomes to support my understanding of the analyses that are being conducted. While it is possible with further explanation that these may not have impacted the results, or perhaps have already been accounted for, without this I cannot interpret these results in a meaningful way. I basically stopped my review on page 6 to allow for a response before attempting to review the remainder of the paper.

Major comments

- Page 6, paragraph 1: To what extent might the depletion of genetic support be explained by the confounding between high rates of failure of oncology, over-representation of oncology in the data, and the dearth of germline genetic evidence for oncology mechanisms? You could be seeing Simpson's paradox here. More careful stratified analysis is needed here. Further, based on Stable 5, you only have four successful trials in the dataset you are analyzing. How can you conduct a reasonable assessment of effects on stopping without successful trials? I am confused at exactly what analysis is being carried out here. Are you conducting analyses with data you haven't described or provided in the supplementary tables?
- There isn't sufficient information in Stable 5 to allow the reader to reproduce your analyses to test for effects like these themselves.
- Sfigure 3: How do you explain that such a large fraction of all trials were terminated due to Covid-19? Is this a subset of all trials? Is the fraction of all active trials heavily weighted towards that last few years? Do you limit to trials that ended within the last few years? Some contextualization and understanding of the temporality of this would help.

Minor comments

- Introduction
 - 1st paragraph, "While there are myriad": this sentence is difficult to parse and the meaning is unclear. What does the 79% refer to? Is that they are major and expensive a source of attrition? How do unforeseen safety issues reflect an unsupported therapeutic hypothesis? There are a wide range of biological reasons for safety risks, many unrelated to the therapeutic hypothesis. I suggest rethinking and rewriting this sentence.
 - "Recently, support from genetic studies" should be clarified, as genetic studies can include a wide range of in vitro and in vivo experiments. These references specifically refer to supporting human genetic evidence.
 - 2nd paragraph, "Clearly, ...": this appears to be a partial thought. How does overall success rates impact positive trial reporting or vice versa. What is the disconnect? Consider revising.
 - "Having access to negative results..." Previously in this paragraph you are talking about trial publications. This sentence is more about a need or opportunities for more fully annotated trial databases. The next two sentences don't carry this forward, but are more like stand-alone statements. I suggest rethinking the arc of this paragraph and reworking.
- Results
 - Sfigure 2: How did you select these reasons for display? Are they the most frequent? While the safety and negative are directionally consistent with what we would expect, are these differences statistically significant? What other reasons showed phase-dependent effects?
 - Page 5: "Of all the studies with a submitted stop reason". This sentence is unclear. What is 48% an overrepresentation of? What is the expectation and how did you arrive at that? Why does recent inclusion of stop reasons bias in favor of oncology? I can only guess from the presented information.
- Supplement
 - Page 2, paragraph 2, note the typo: Another reason for why certain classes from different meta-classes are semantically similar may [be] the multiclass nature of the dataset
 - Sfigure 1: Should add note to the caption that 447 Covid-19 terminations were added to the Pak data set to avoid confusion with the original data set (i.e. not available in Stable 1). Shouldn't the additional Covid-19 failures be added to Stable 1? Without them it's not clear that the analyses in this paper could be reproduced.

Sincerely,

Matt Nelson
VP, Genetics & Genomics
Deerfield

Reviewer #2:

Remarks to the Author:

This is a well-written paper that is novel its use of clinical trial stopping reasons to assess variables associated with successful drug targets. I believe it is of interest to a large audience interested in the application of genetics to drug discovery and development.

The authors trained a NLP model to categorize stopping reasons for clinical trials that terminate before an endpoint is reached. They determine other variables that are correlated with stopping reason, focusing on genetics and genomics data sources. They find that clinical trials that have prematurely stopped are less likely to have genetic support, and that the effect is largest for those that terminated for efficacy reasons.

The authors' annotations of clinical trial stopping reasons from clinicaltrials.gov will be valuable for researchers working with clinical trial data at scale and using trial outcomes to train machine learning models. This need not be limited to researchers in genetics or genomics; other areas of study related to drug development may also find such a dataset to be useful. Identifying current clinical and approved drugs and their MOAs is a relatively straightforward way to construct a positive set for machine learning approaches, but understanding what drugs or MOAs should belong to a negative set is a common challenge to machine learning methods in this field. For example, it is not uncommon for an ultimately successful MOA to be preceded by unsuccessful trials using drugs with the same MOA. There is an intuition that understanding the reasons for clinical trial failure or termination should help separate failures due an ineffective therapeutic target from other failures, as well as to help identify failed clinical drugs with good safety profiles that could be repurposed in other indications, but this has not been tested or applied at scale to my knowledge.

Of the findings on variables that are associated with clinical trial stopping reasons, I find those associated with safety to be particularly compelling (Fig 3). To my knowledge, tissue-specific expression has been linked to successful drug targets, but evidence on genetic constraint has been more mixed. Separating safety failures from other reasons may help make sense of the observed patterns.

I believe the paper is well-written, and the methodology applied is justified and adequately documented. My main struggle with this and other similar work is how to interpret results of single variable analyses in light of the numerous, sometimes correlated variables impacting trial outcomes. Multivariate analysis is one possibility though the interpretation of coefficients from a multivariate regression has its own set of challenges.

The authors are to be commended for their transparency of including data for the full set of comparisons between clinical trial outcome variables and predictor variables. I do have a specific concern Supp Table 6. The authors have negative values for the column "d" for several entries. Assuming a, b, c, d are cell counts for a 2x2 table this should not occur. Also, $a+b+c+d$ does not always sum to total and I am not sure if that is expected. I would request that if my interpretation of

these columns is correct that the authors check their calculations.

Assuming the results remain constant after any concerns about the calculations are addressed, I don't doubt that most of the reported significant associations are genuine. However, the large number of highly significant associations makes it difficult to understand which variables are the most important, and how correlations between predictor variables are affecting the associations. In particular, I wonder if the large differences between oncology and non-oncology trials in terms of stopping reasons could explain some other associations that otherwise may seem counterintuitive. In contrast to germline and mouse genetic evidence, somatic evidence appears negatively associated with Phase II-IV clinical trials as a whole, but positively associated with safety failures. This seems entirely consistent with the fact that cancer drugs are more likely to have somatic evidence and also are more likely to fail early in their development for safety reasons. Would it be possible to stratify some of these results by oncology and non-oncology indications?

It is clearly outside the scope of this work to pursue this in detail, but I wonder if the authors have thoughts about the value and possibility (or lack thereof) of integrating the data here with outcomes from clinical trials that were not prematurely stopped, but the assets were not pursued further in the indication. Is there or could there be a comparable data source with similar annotations of why the asset did not progress? Many broad categories of reason for premature stopping could also apply here (e.g. business, efficacy).

Minor comments:

- The link <https://huggingface.co/datasets/opentargets> does not work. It appears the correct link may be https://huggingface.co/opentargets/clinical_trial_stop_reasons.

Reviewer #3:

Remarks to the Author:

Razuvayevskaya et al. applied NLP to classify 28,842 clinical trials for reasons why they stopped before endpoints were met. They examined these classes based on underlying evidence and target properties and show that trials are more likely to stop due to lack of efficacy in absence of strong genetic support. Also, the study demonstrates trials are likely to stop for safety reasons for drug target genes that highly constrained or not selectively expressed.

Although studies have shown the importance of human (and mouse) genetic data to support clinical trial outcomes, these previous studies have focused on successful clinical trial progression. However, studies have not focused on why clinical trials fail in large part because of difficulty in obtaining the free text reasons for failure in drug datasets. This study overcomes this by using NLP to classify stopping reasons from clinicaltrials.gov and then examine genetic evidence from the Open Targets platform.

Overall, this is an important study that adds to the current drug genetics literature on using human genetics to de-risk negative results of clinical trials.

My comments are below:

1. The methods section is very short and relies / points to a previous paper for key methodological

details about the different sources of genetic support, clinical trial outcome datasets, also how the phenotypes between genetic and clinical trial data were mapped. Rather than having to go back to that paper, these methods should be abbreviated / summarized in the main manuscript and/or supplementary materials.

2. Also, please provide more details about the association analysis e.g. how were repeating observations such as multiple gene targets for a single drug, multiple drugs for the same clinical trial phenotype handled in the association analysis for genetic evidence and clinical trial outcomes (e.g. shown in Figure 2).

3. A main finding is that clinical trials that stopped due to a negative reason showed a significant decrease of genetic support (OR=0.61, P=6E-18). This is a nice result and supports what we would expect.

The finding of other prediction reasons for stopping a trial such as insufficient enrollment, problems with study design, or business or administration reasons showing strong to moderate depletion of genetic evidence, is a bit less clear and expected, at least to me. The authors suggest this observation denotes "potential reduced support for the therapeutic hypothesis". However, I would have thought these other reasons are logistical reasons (e.g. study design issues, business / admin, insufficient enrollment) and not biological reasons that would support a genetic evidence / clinical trial stoppage relationship. I was somewhat expecting these other reasons to serve as a negative control similar to the COVID-19 null association. Can the authors provide more explanation, clarity for why they think these results are likely explained by a weak therapeutic hypothesis?

4. Following up on the above comment, I'm wondering about whether additional negative control analyses can be performed. The COVID-19 null association is the lone one but even though its not significant, the 95% error bars are large because of the small N. Some suggestions for negative control analyses:

A) Randomly permute the drug target genes and run 100 permutations and then perform genetic evidence and clinical trial outcome association in Figure 2. We expect only 5% of associations to be significant using nominal $P < 0.05$ cutoff.

B) The study requires mapping the phenotypes for the genetic evidence with the same phenotype for the clinical trial. Consider changing the mapping scheme so that the genetic evidence phenotype does NOT map to the same phenotype in the clinical trial but instead maps to a completely unrelated clinical trial phenotype. Perform same analysis in Figure 2 and again we expect only 5% of associations to be significant using nominal $P < 0.05$ cutoff.

Minor:

1. Introduction: Duffy et al. 2020 found that constrained genes were associated with increased side effects (not clinical trial stoppage)

2. Figure 2 / 3 (and all other figures/tables that apply): I suggest providing the actual P-values rather than using the starred binning system. This provides more exact information about statistical significance of the association enrichments.

3. Supp. Figure 4 - its a bit hard to assess the OR, 95% CI and P-values for the different types of genetic support by looking at the plot. Can the authors please provide these in a Supp. Table.

4. typo: enrolment

5. Figure 3 legend - the cutoffs used to define some the classes in the categories are needed and should be defined somewhere either in the figure legend, methods or supplementary materials. e.g. very high, high, medium, etc. for genetic constraint; different classes for RNA specificity

Author Rebuttal to Initial comments

Response to referees

Please find next a point-by-point response to the reviewer's comments. We thank all the reviewers for the time spent reviewing the manuscript and for their constructive feedback. We believe the quality of the revised manuscript is significantly improved compared to the original version in large due to their scientific input.

Dear Dr Ochoa,

Your Article, "Why Clinical Trials Stop: The Role of Genetics" has now been seen by 3 referees. You will see from their comments below that while they find your work of interest, some important points are raised. We are interested in the possibility of publishing your study in Nature Genetics, but would like to consider your response to these concerns in the form of a revised manuscript before we make a final decision on publication.

In brief, the three reviews all acknowledge the potential broad interest in your work, but are split on the overall robustness of your analysis.

Reviewer #1 highlights a major issue: they suggest that the large number of oncology trials is skewing the results of your analysis and that this must be thoroughly investigated before your analysis can be reviewed in its entirety.

We note that Referee #2 also comments on the preponderance of oncology trials - suggesting this is an especially important issue to address in a revision - but otherwise sounds quite positive.

Reviewer #3 is also, in the main, positive and has also makes a few suggestions for new analysis.

To guide the scope of the revisions, the editors discuss the referee reports in detail within the team, including with the chief editor, with a view to identifying key priorities that should be addressed in revision and sometimes overruling referee requests that are deemed beyond the scope of the current study. We hope that you will find the prioritized set of referee points to be useful when revising your study. Please do not hesitate to get in touch if you would like to discuss these issues further.

We therefore invite you to revise your manuscript taking into account all reviewer and editor comments. Please highlight all changes in the manuscript text file. At this stage we will need you to upload a copy of the manuscript in MS Word .docx or similar editable format.

*2) If you have not done so already please begin to revise your manuscript so that it conforms to our Article format instructions, available

here.

*3) Include a revised version of any required Reporting Summary:

Please be aware of our guidelines on digital image standards.

<https://mts-ng.nature.com/cgi-bin/main.plex?el=A1G2BKp6A2Bcjm3J6A9ftdkjhhlzqM6GlbBF0aGjK8wZ>

Sincerely,

Michael Fletcher, PhD

Senior Editor, Nature Genetics

ORCID: 0000-0003-1589-7087

Referee expertise:

Referee #1: genetics; clinical trials; genomics.

Referee #2: genetics/genomics; clinical trials.

Referee #3: genetics, genomics, machine learning.

Reviewers' Comments:

Reviewer #1:

Remarks to the Author:

Razuvayevskaya et al. present the training of a natural language processing model to identify the reason for stopping clinical trials in the large [Clinicaltrials.gov](https://clinicaltrials.gov) database and analysis of some of the genetic and genomic factors that influence those stopping reasons. From the validation exercise carried out by the authors, they have created a valuable model and classified data set that will support other research in this area. Their analyses of this data set may provide valuable insights that can impact drug discovery and development decisions.

Based on the modest amount of methodological description in the paper, I am concerned with the robustness of the conclusions the authors draw from their analyses. My greatest concerns are 1) ignoring the impact that such a large fraction of oncology studies may have on the primary study measures and 2) the lack of positive trial outcomes to support my understanding of the analyses that are being conducted. While it is possible with further explanation that these may not have impacted the results, or perhaps have already been accounted for, without this I cannot interpret these results in a meaningful way. I basically stopped my review on page 6 to allow for a response before attempting to review the remainder of the paper.

We appreciate the positive comments from the reviewer on the value of the produced research as well as its impact on drug discovery and development. We have also taken very seriously the constructive feedback, particularly on the two points raised as discussed below. We hope the reviewer is satisfied with the additional analysis, results and materials included in this response.

Major comments

- Page 6, paragraph 1: To what extent might the depletion of genetic support be explained by the confounding between high rates of failure of oncology, over-representation of oncology in the data, and the dearth of germline genetic evidence for oncology mechanisms? You could be seeing Simpson's paradox here. More careful stratified analysis is needed here.

As discussed in the manuscript and reiterated by the reviewer, the preponderance of oncology studies during the last 15 years, as well as their unique genetic basis could confound some of the results presented in this manuscript. To further investigate the impact of oncology trials, we repeated the analysis by splitting the clinical trial indications into "all indications", "oncology" or "non-oncology". We evaluated every case-control study interrogating the association between the presence of genetic evidence and the predicted reason for stopping the trial.

New Supplementary Figure 6. Genetic support for stopped clinical trials in all indications, non-oncology and oncology. The panels show the odds ratio of support for the target-disease hypothesis from genetics evidence for stopped clinical trials split by higher level stopping reason (bottom row). Significant odds ratios > 1 are enriched and <1 are depleted for genetic evidence in each subclass of trial. The number of clinical studies genetically supported and stopped due to the reason (n), Odds ratio (OR), 95% confidence intervals (CI) and Fisher exact test p-values are displayed in the columns.

The results added in the new Supplementary Figure 6 confirm that studies stopped due to “Negative” reasons are depleted in genetic evidence for Non-oncology studies (OR 0.75, $p = 0.00026$) and oncology studies (OR 0.53, $p = 2.5e-13$). As a consequence of stratifying the studies, some tests are less powered and should be taken with additional caution. Therefore, we decided not to provide further stratification of the non-oncology indications. In light of these additional results, we believe the results are reasonably consistent across indications, most likely not suffering from Simpson’s paradox, but we would be happy to perform additional validations if any concerns emerge.

Additionally, we also stratified by the same indication groupings the analysis of the factors associated with studies stopped due to safety or side effects (new Supplementary Figure 8). Although all the effects observed when aggregating all indications could be reproduced in the oncology indications, we couldn’t replicate the statistical significance for the non-oncology indications. As reported in the new Supplementary Figure 8, the odds generally agreed with the ones observed at the aggregate level, but we didn’t observe statistical significance probably due to the reduced power. For this reason, we clarified in the main text that the signals observed in this section were largely driven by studies stopped due to safety concerns in oncology.

New Supplementary Figure 8. Risk of stopping a clinical trial due to safety or side effects in all indications, oncology and non-oncology. Odds ratio (OR) > 1 represents an increased risk of study stopping and OR < 1 protection against stopping. Clinical trials are split by therapeutic area, relative genetic constraint of the target (classified into 6 groups using the respective percentiles as defined by GnomAD), intolerance to loss of function (LOF) mutations (GnomAD), tissue specificity (in groups defined by HPAs criteria) and by network connectivity (IntAct).

Further, based on Stable 5, you only have four successful trials in the dataset you are analyzing How can you conduct a reasonable assessment of effects on stopping without successful trials? I am confused at exactly what analysis is being carried out here. Are you conducting analyses with data you haven't described or provided in the supplementary tables?

The “successful” class in the stop reason classification (now Supplementary Table 6) refers to the cause of stopping the trial and not to the outcome of the study. All the stopped studies due to the “successful” class have stopped due to some positive results. The gold standard from Pak *et al.* (Supplementary Table 1) contains 7 such cases:

- “The incidence of post-operative delirium observed from interim blinded data was significantly lower than the current literature in this population.”
- “The incidence of post-operative delirium observed from interim blinded data in DEX-06-09 was significantly lower than the current literature in this population.”
- “The data available is enough for the publication and the study article published in Middle East Journal of Anesthesiology.”
- “survival advantage demonstrated”
- “Clearly identifiable benefits 50% of patients included”
- “Interim analysis showed a significant reduction in the pain scores”
- “Results showed statistically significant benefit in the experimental group”

Unfortunately, due to the heterogeneous nature of the language and the small gold standard, the model underperformed when classifying the “Success” category as described in Supplementary Table

3 and Supplementary Table 4. Considering how infrequent this class was and the poor quality of the model, this class was ignored in the analysis aimed at interpreting the causes behind clinical trial stoppage. We understand the confusion from the reviewer and we have significantly enhanced the supplementary methods to clarify any potential misunderstandings.

• There isn't sufficient information in Stable 5 to allow the reader to reproduce your analyses to test for effects like these themselves.

To make the analysis clearer, we have included Supplementary Figure 4 to reflect the workflow of data and analysis performed including the most relevant counts described the data landscape.

New Supplementary Figure 4: Data flow leading universe of clinically investigated target-disease pairs.

Data sources, steps and filters leading to the universe of clinical studies used for case-control studies (see Supplementary Methods for more information).

Also based on the reviewer's comment, we recognised the code that downloads the datasets and performs the analysis might not be transparent to all readers. Therefore, we added a new Supplementary Table 8 that captures all the annotations required to perform the case-control analysis as described in the expanded supplementary methods. The result of all individual tests together with the contingency tables (a,b,c,d) remains available in Supplementary Table 7 (former Supplementary Table 6).

• Sfigure 3: How do you explain that such a large fraction of all trials were terminated due to Covid-19? Is this a subset of all trials?

As reported in Supplementary Table 2 and mentioned in the main text, we curated 460 trials stopped due to COVID-19. This process involved a simple pattern-matching of several keywords (COVID, pandemic, Sars-Cov, etc.) and further manual curation of the resulting list. The model trained using the manual curation predicted 1,421 studies stopped due to reasons that included the pandemic (Supplementary Table 5). This figure corresponds to 5% of all the stopped trials predicted. In supplementary figure 3, percentages are reported by therapeutic area therefore the overall 5% might vary depending on the incidence of COVID-19 in each of the therapeutic areas. We amended the figure caption to clarify this is a fraction of all the trials in the therapeutic area and also to include the overall incidence of the referred classes across all TAs.

- Is the fraction of all active trials heavily weighted towards that last few years? Do you limit to trials that ended within the last few years? Some contextualization and understanding of the temporality of this would help.

In Figure 1, we aimed to provide a temporal context by representing every stopped trial based on the year the study started. The start date was the most complete and consistent time reference from all the available dates in ClinicalTrials.gov, although it might be several years apart from when the study was stopped. The next plot illustrates the observed trend in the number of studies with stopped reason.

Stopped studies submitted to ClinicalTrials.gov by start date.

As expected, the availability of studies is influenced by the availability of submissions providing a “why stopped” reason on ClinicalTrials.gov. This field was made available in ClinicalTrials.gov in 2008 and the number of submitted studies has grown significantly since Pak et al. performed the manual curation on May 31, 2010. Considering the trends in the pharmaceutical industry during the last 15 years, this data confirms the large presence of oncology studies in the analysed dataset.

Minor comments

- Introduction

- 1st paragraph, "While there are myriad": this sentence is difficult to parse and the meaning is unclear. What does the 79% refer to? Is that they are major and expensive a source of attrition? How do unforeseen safety issues reflect an unsupported therapeutic hypothesis? There are a wide range of biological reasons for safety risks, many unrelated to the therapeutic hypothesis. I suggest rethinking and rewriting this sentence.

We rewrote this section to increase clarity. We also removed the therapeutic hypothesis concept, which can cause misinterpretation without further explanation.

- "Recently, support from genetic studies" should be clarified, as genetic studies can include a wide range of in vitro and in vivo experiments. These references specifically refer to supporting human genetic evidence.

The sentence was rewritten to increase readability as suggested. We also included references to new literature that emerged since the first submission of this draft such as the recent Nature Review: "From target discovery to clinical drug development with human genetics" from Trajanoska *et al.* and the medrxiv preprint by Vallabh Minikel *et al.* "Refining the impact of genetic evidence on clinical success" in which the reviewer contributed as an author.

- 2nd paragraph, "Clearly, ...": this appears to be a partial thought. How does overall success rates impact positive trial reporting or vice versa. What is the disconnect? Consider revising.

Indeed, it was a partial thought. The paragraph has been rewritten to increase clarity.

- "Having access to negative results..." Previously in this paragraph you are talking about trial publications. This sentence is more about a need or opportunities for more fully annotated trial databases. The next two sentences don't carry this forward, but are more like stand-alone statements. I suggest rethinking the arc of this paragraph and reworking.

The paragraph has been reworked to enhance the narrative, as suggested by the reviewer.

• Results

- Sfigure 2: How did you select these reasons for display? Are they the most frequent? While the safety and negative are directionally consistent with what we would expect, are these differences statistically significant?

While we recognise Supplementary Figure 2 contained the necessary data to illustrate some patterns in the data, we share the reviewer's point we could benefit from a more rigorous and systematic analysis on the patterns emerging from the relationships between the clinical phases and the stopped studies. For this reason, we performed a test for every phase and reason and presented it in the new Supplementary Figure X, only excluding classes with a low number of studies ("Success", "Insufficient_Data", "No_Context", "Interim_Analysis", "Another_Study").

New Supplementary Figure 2: Association between the clinical phase and the reason for stopping the study. The analysis considers all predicted stop reasons and their clinical phase as reported by ClinicalTrials.gov.

As described in the main text and corroborated by the new analysis, certain stop reasons can be statistically associated with specific clinical phases, largely aligning with the purpose of each phase. We could reproduce the three observations reported in the main text: study or staff moved more predominant in early phase I, safety or side effects mainly affecting phases I and II and negative reasons peaking their impact in phase III. With the additional tests, we can inform about the directionality as well as the significance as requested by the reviewer.

o Page 5: "Of all the studies with a submitted stop reason". This sentence is unclear. What is 48% an overrepresentation of? What is the expectation and how did you arrive at that? Why does recent inclusion of stop reasons bias in favor of oncology? I can only guess from the presented information.

This paragraph has been rewritten according to the comments from the reviewer. 48% of the stopped trials belong to oncology indications which is a much larger representation than the second most frequent therapeutic area, neurological diseases with 9%. We explain now in the main text, how this large percentage is compatible with previous observations. More precisely, we refer to oncology being the most frequent therapeutic area for recent approvals - 27% of 2022 approvals (Mullard 2023) - and also oncology having the largest incidence of failures (32%) among all the therapy areas (Harrison 2016). These 2 figures combined would generate a figure similar to the one observed in a similar period.

• Supplement

o Page 2, paragraph 2, note the typo: Another reason for why certain classes from different metaclasses are semantically similar may [be] the multiclass nature of the dataset

The typo was addressed in the supplementary material.

○ Sfigure 1: Should add note to the caption that 447 Covid-19 terminations were added to the Pak data set to avoid confusion with the original data set (i.e. not available in Stable 1). Shouldn't the additional Covid-19 failures be added to Stable 1? Without them it's not clear that the analyses in this paper could be reproduced.

The caption for Supplementary Figure 1 was updated to include the consideration of the additional 447 stopped trials due to COVID-19. Regarding supplementary table 1, we wanted to preserve the curation of Pak *et al.* as this was never made publicly available. For this reason, we decided to include the additional curation of studies stopped due to COVID-19 as Supplementary Table 2. Hopefully, this was clear in the main text: “Moreover, we added 447 studies stopped due to the COVID-19 pandemic (Supplementary Table 2).”. This data should be sufficient to reproduce the training of the model. Moreover, the training dataset is also available in the HuggingFace Hub where at the time of writing this response had more than 200 downloads in the last month.

Sincerely,

Matt Nelson

VP, Genetics & Genomics

Deerfield

Reviewer #2:

Remarks to the Author:

This is a well-written paper that is novel its use of clinical trial stopping reasons to assess variables associated with successful drug targets. I believe it is of interest to a large audience interested in the application of genetics to drug discovery and development.

We appreciate the comments and the time spent to review the manuscript.

The authors trained a NLP model to categorize stopping reasons for clinical trials that terminate before an endpoint is reached. They determine other variables that are correlated with stopping reason, focusing on genetics and genomics data sources. They find that clinical trials that have prematurely stopped are less likely to have genetic support, and that the effect is largest for those that terminated for efficacy reasons.

The authors' annotations of clinical trial stopping reasons from clinicaltrials.gov will be valuable for researchers working with clinical trial data at scale and using trial outcomes to train machine learning models. This need not be limited to researchers in genetics or genomics; other areas of study related to drug development may also find such a dataset to be useful. Identifying current clinical and approved drugs and their MOAs is a relatively straightforward way to construct a positive set for machine learning approaches, but understanding what drugs or MOAs should belong to a negative set is a common challenge to machine learning methods in this field. For example, it is not uncommon for an ultimately successful MOA to be preceded by unsuccessful trials using drugs with the same MOA. There is an intuition that understanding the reasons for clinical trial failure or termination should help separate failures due an ineffective therapeutic target from other failures, as well as to help identify failed clinical drugs with good safety profiles that could be repurposed in other indications, but this has not been tested or applied at scale to my knowledge.

We sincerely thank the reviewer for the discussion on the impact of this research and the value of interpreting the many causes that might lead to failure, as well as its implications in drug discovery and development.

Of the findings on variables that are associated with clinical trial stopping reasons, I find those associated with safety to be particularly compelling (Fig 3). To my knowledge, tissue-specific expression has been linked to successful drug targets, but evidence on genetic constraint has been more mixed. Separating safety failures from other reasons may help make sense of the observed patterns.

We agree with the reviewer the same factor (e.g. genetic constraint) could or could not be informative of success, but it's a different question whether it could be informative to one of the many factors that can lead to failure, in this case, safety or adverse events. This is the question we aimed to address in this study. As described in the response to Reviewer #1, we are confident, that genetic constraint is indicative of safety risks in the overall portfolio, but when looking by therapeutic areas we can only say conclusively that it's an important factor to consider in oncology studies. Despite the promising odds we are not statistically supported to generalise this statement to other indications. We have amended the manuscript accordingly. As pointed out by the reviewer, previous reports indicated that drug targets were on average just slightly more constrained than all genes when looking at all indications (Vallabh Minikel *et al.* Nature 2020). We now report the differences in genetic constraints between failures and ongoing clinical studies or approved drugs.

I believe the paper is well-written, and the methodology applied is justified and adequately documented. My main struggle with this and other similar work is how to interpret results of single variable analyses in light of the numerous, sometimes correlated variables impacting trial outcomes. Multivariate analysis is one possibility though the interpretation of coefficients from a multivariate regression has its own set of challenges.

We thank the reviewer for the comments about the appropriate methodology and documentation. However, we agree with reviewer #1 and #3 that some methodological description was incomplete to fully reproduce the study. We have tried our best to complete the gaps and achieve the best possible standard.

Regarding the analysis, we agree with the reviewer that multivariate analysis represents a powerful methodology that comes with its challenges. The main reason why we were discouraged from using it in this study was the limited data we had to test some of our hypotheses. The sparse data on some of our features prevented us from performing rigorous tests. However, as more clinical failures accumulate and the wealth of human genetic evidence increases, we hope we could benefit from meaningful coefficients we could learn from. We will certainly explore this in our future research.

The authors are to be commended for their transparency of including data for the full set of comparisons between clinical trial outcome variables and predictor variables. I do have a specific concern Supp Table 6. The authors have negative values for the column "d" for several entries. Assuming a, b, c, d are cell counts for a 2x2 table this should not occur. Also, $a+b+c+d$ does not always sum to total and I am not sure if that is expected. I would request that if my interpretation of these columns is correct that the authors check their calculations.

As raised by the reviewers, we have increased the transparency of the study by including more context on how the information retrieved from public sources is processed in the study. We have included new supplementary methods, tables and figures to make intermediate results more accessible to the readers of Nature Genetics. In particular, we have included Supplementary Table Supplementary Table 8 which contains the full set of comparisons between clinical trial outcome variables and predictor variables as requested by the reviewer.

As well spotted by the reviewer, the contingency tables had an issue in their calculation that led to some cells presenting negative values on rare occasions. This error has been addressed and all calculations and figures re-computed based on the new estimates. Supplementary Table 6 (now ST7)

was also updated accordingly. Although the results and figures were updated, none of the findings in this study were affected by the issue, so the conclusions remain the same.

Assuming the results remain constant after any concerns about the calculations are addressed, I don't doubt that most of the reported significant associations are genuine. However, the large number of highly significant associations makes it difficult to understand which variables are the most important, and how correlations between predictor variables are affecting the associations. In particular, I wonder if the large differences between oncology and non-oncology trials in terms of stopping reasons could explain some other associations that otherwise may seem counterintuitive. In contrast to germline and mouse genetic evidence, somatic evidence appears negatively associated with Phase II-IV clinical trials as a whole, but positively associated with safety failures. This seems entirely consistent with the fact that cancer drugs are more likely to have somatic evidence and also are more likely to fail early in their development for safety reasons. Would it be possible to stratify some of these results by oncology and non-oncology indications?

Hopefully, the separation of oncology versus non-oncology comparison described in the response to reviewer #1 will complement the previous results and satisfy the reviewer.

It is clearly outside the scope of this work to pursue this in detail, but I wonder if the authors have thoughts about the value and possibility (or lack thereof) of integrating the data here with outcomes from clinical trials that were not prematurely stopped, but the assets were not pursued further in the indication. Is there or could there be a comparable data source with similar annotations of why the asset did not progress? Many broad categories of reason for premature stopping could also apply here (e.g. business, efficacy).

The reviewer brings up an interesting topic about other types of negative outcomes not captured by prematurely stopped trials. As reported in this study, stopped trials (withdrawn, terminated and suspended) only represent <10% of the clinical trials on ClinicalTrials.gov, but a large fraction of the remaining 90% led to unsatisfactory results that could be catalogued as a failure, occasionally leading to the closure of the program. We agree there is a lack of understanding about these results which usually represent a delta between the expectation or the regulatory requirements and the observed results. This more subtle difference is - as far as we know - not concisely summarised in the same way that stopped studies are. Thus, stricter submission systems or more sophisticated systematic post-processing of the trial results would be required to systematically understand what is considered unsatisfactory in each of these completed studies, or what other external decisions influenced the discontinuation of the program. Aggregate numbers from analysis from commercial companies are occasionally reported (Dowden and Munro NRDD 2019), but - to our knowledge - there are no publicly available resources that capture such outcomes.

Minor comments:

- The link <https://huggingface.co/datasets/opentargets> does not work. It appears the correct link may be https://huggingface.co/opentargets/clinical_trial_stop_reasons.

Indeed the link pointed to an obsoleted page. This issue has been addressed including a link to the trained model in Hugging Face, as well as the full training set that was used to create the model. Moreover, the model can now interactively be queried through the Hugging Face website to assist any scientist interested in classifying their reasons.

Reviewer #3:

Remarks to the Author:

Razuvayevskaya et al. applied NLP to classify 28,842 clinical trials for reasons why they stopped before endpoints were met. They examined these classes based on underlying evidence and target properties and show that trials are more likely to stop due to lack of efficacy in absence of strong genetic support. Also, the study demonstrates trials are likely to stop for safety reasons for drug target genes that highly constrained or not selectively expressed.

Although studies have shown the importance of human (and mouse) genetic data to support clinical trial outcomes, these previous studies have focused on successful clinical trial progression. However, studies have not focused on why clinical trials fail in large part because of difficulty in obtaining the free text reasons for failure in drug datasets. This study overcomes this by using NLP to classify stopping reasons from clinicaltrials.gov and then examine genetic evidence from the Open Targets platform.

Overall, this is an important study that adds to the current drug genetics literature on using human genetics to de-risk negative results of clinical trials.

My comments are below:

1. The methods section is very short and relies / points to a previous paper for key methodological details about the different sources of genetic support, clinical trial outcome datasets, also how the phenotypes between genetic and clinical trial data were mapped. Rather than having to go back to that paper, these methods should be abbreviated / summarized in the main manuscript and/or supplementary materials.

As requested by all three reviewers and described in previous responses, we have expanded the Supplementary Methods, Supplementary Tables and Supplementary Figures to include a detailed description of our data, methodology and results. We aimed to make more explicit all the analysis and intermediate outputs hoping to help any future readers of the article.

2. Also, please provide more details about the association analysis e.g. how were repeating observations such as multiple gene targets for a single drug, multiple drugs for the same clinical trial phenotype handled in the association analysis for genetic evidence and clinical trial outcomes (e.g. shown in Figure 2).

The Supplementary Methods section has been expanded to include a detailed description of how the case-control studies were performed and how we derived statistical relationships for all combinations of exposures and outcomes.

3. A main finding is that clinical trials that stopped due to a negative reason showed a significant decrease of genetic support (OR=0.61, P=6E-18). This is a nice result and supports what we would expect.

The finding of other prediction reasons for stopping a trial such as insufficient enrollment, problems with study design, or business or administration reasons showing strong to moderate depletion of genetic evidence, is a bit less clear and expected, at least to me. The authors suggest this observation denotes “potential reduced support for the therapeutic hypothesis”. However, I would have thought these other reasons are logistical reasons (e.g. study design issues, business / admin, insufficient enrollment) and not biological reasons that would support a genetic evidence / clinical trial stoppage relationship. I was somewhat expecting these other reasons to serve as a negative control similar to the COVID-19 null association. Can the authors provide more explanation, clarity for why they think these results are likely explained by a weak therapeutic hypothesis?

As commented in the discussion, we were similarly puzzled by some of these less-obvious associations in which reasons that we anticipated to be independent of biology, presented a weaker genetic support. Our interpretation is that the decision to halt a clinical study is probably influenced by multiple factors. The FDA mandate to provide a reason to stop the study in ClinicalTrials.gov might potentially bias submitters to provide one of the many reasons why the study was stopped. In this sense, we expect the COVID-19 pandemic reason to represent a much cleaner negative control, as we anticipate many of these trials had to be stopped independently of how strong the biological hypothesis was in the first place.

4. Following up on the above comment, I'm wondering about whether additional negative control analyses can be performed. The COVID-19 null association is the lone one but even though its not significant, the 95% error bars are large because of the small N. Some suggestions for negative control analyses:

A) Randomly permute the drug target genes and run 100 permutations and then perform genetic evidence and clinical trial outcome association in Figure 2. We expect only 5% of associations to be significant using nominal P<0.05 cutoff.

B) The study requires mapping the phenotypes for the genetic evidence with the same phenotype for the clinical trial. Consider changing the mapping scheme so that the genetic evidence phenotype does NOT map to the same phenotype in the clinical trial but instead maps to a completely unrelated clinical trial phenotype. Perform same analysis in Figure 2 and again we expect only 5% of associations to be significant using nominal P<0.05 cutoff.

The definition of appropriate negative controls is indeed a complex topic. Sparsity and the lack of independence complicate any rigorous statistical assessment. To the best of our knowledge, we attempted to validate reviewer's suggestion A, by conducting 100 permutations of the genetic evidence dataset and estimating the associations with all predicted reasons for stopping a trial.

Association between stop reasons and 100 random permutations of the genetic support dataset. “n” values represent the mean size of intersections between the stopped clinical studies with the given reason and the random genetic support.

The enrichment signals predominantly centre around an Odds Ratio of 1 ($\log OR \sim 0$) except for poorly powered classes such as “Success” or “Insufficient data”. One of the limitations of the performed analysis is linked to the sparsity of the data. The randomisation of the genetic data reduces the chances of finding overlapping clinical studies, occasionally producing estimates based on a very low number of overlaps. For example, the mean number of overlaps in the class “Insufficient enrollment” across the 100 permutations is 225, whereas the actual estimate shown in Figure 2 is based on 947 observations.

Another problem is derived from the dependency of clinical studies. Although we randomised the genetic support, multiple stopped clinical trials might target the same gene-disease hypothesis. This lack of independence might artificially create large effect sizes when a random genetically supported gene disease overlaps a group of clinical studies. The next QQ plot illustrates the “Business or administrative” reason when comparing the observed p-values (through permutation) against those expected in a normal distribution.

QQ plot derived from 100 permutations of the “Business or administrative” reason. Observed p-values are derived from Fisher’s exact test when analysing 100 permutations of the genetic association dataset and the gene-disease associations extracted from the clinical trials dataset.

Overall, we found no concern but there are several caveats in the analysis performed that prevented us from deriving additional findings. We would be happy to investigate further if there are any additional concerns.

Minor:

1. Introduction: Duffy et al. 2020 found that constrained genes were associated with increased side effects (not clinical trial stoppage)

This incorrect statement and discussed in the results with the appropriate reference.

2. Figure 2 / 3 (and all other figures/tables that apply): I suggest providing the actual P-values rather than using the starred binning system. This provides more exact information about statistical significance of the association enrichments.

Figure 2 and Figure 3 now include the p-values instead of the binned representation using stars as suggested by the reviewer. All statistical tests are also available in Supplementary Table 7 (previous ST6).

3. Supp. Figure 4 - its a bit hard to assess the OR, 95% CI and P-values for the different types of genetic support by looking at the plot. Can the authors please provide these in a Supp. Table.

Considering the difficulties of the reviewer to properly visualise the figure, we decided to represent the same information with a different layout that hopefully satisfies the reviewer and also the readers. The figure legend was also adjusted accordingly to enhance readability. Moreover, odds ratio, confidence intervals and p-values for all statistical tests included in the manuscript are available in Supplementary Table 7 (previous ST6). Since the analysis focuses on comparing the enrichments by each of the data sources, the relevant rows are those that contain the label “byDatasource” in the “comparisonType” column.

4. typo: enrolment

For consistency with ClinicalTrials.gov, we adopted the American spelling “enrollment” as opposed to the British spelling “enrolment” as suggested by the reviewer.

5. Figure 3 legend - the cutoffs used to define some the classes in the categories are needed and should be defined somewhere either in the figure legend, methods or supplementary materials. e.g. very high, high, medium, etc. for genetic constraint; different classes for RNA specificity

All categories in Figure 3 and corresponding supplementary materials are now properly described, as well as in the Supplementary Methods. We appreciate the feedback from the reviewer.

Decision Letter, first revision:

13th Nov 2023

Dear David,

Your Article, "Why Clinical Trials Stop: The Role of Genetics" has now been seen by 3 referees. You will see from their comments below that while they find your work of interest, some important points are raised. We are interested in the possibility of publishing your study in Nature Genetics, but would like to consider your response to these concerns in the form of a revised manuscript before we make a final decision on publication.

In brief, Reviewers #2 and #3 have no further major concerns and support publication.

Conversely, the previously-negative Referee #1 - while appreciating the improvement in revision - thinks that the definition of genetic support used needs to be carefully considered, as this will affect the interpretation of your analysis' results.

We think this is an addressable issue, but given its centrality to your work, we concluded that we would like a full response that we will almost certainly send back to the referee for a (hopefully!) final approval.

To guide the scope of the revisions, the editors discuss the referee reports in detail within the team, including with the chief editor, with a view to identifying key priorities that should be addressed in revision and sometimes overruling referee requests that are deemed beyond the scope of the current study. We hope that you will find the prioritized set of referee points to be useful when revising your study. Please do not hesitate to get in touch if you would like to discuss these issues further.

We therefore invite you to revise your manuscript taking into account all reviewer and editor comments. Please highlight all changes in the manuscript text file. At this stage we will need you to upload a copy of the manuscript in MS Word .docx or similar editable format.

*2) If you have not done so already please begin to revise your manuscript so that it conforms to our

Article format instructions, available here.

*3) Include a revised version of any required Reporting Summary:

Please be aware of our guidelines on digital image standards.

[redacted]

We hope to receive your revised manuscript within four to eight weeks. If you cannot send it within this time, please let us know.

Sincerely,

Michael Fletcher, PhD
Senior Editor, Nature Genetics

ORCID: 0000-0003-1589-7087

Reviewers' Comments:

Reviewer #1:

Remarks to the Author:

I appreciate the additional work the authors have done to address the questions and recommendations provided by me and the other reviewers. The manuscript is substantially improved and it is much easier to follow/dissect the study. While I am still supportive of this work, I have a few additional concerns with how the study was conducted or communicated that should be considered.

Major comments

The definition of genetic support appears to be very broad, which concerns me that the associations being observed are more of a reflection that genes for terminated programs are just more likely to have genetic associations of some sort, rather than some correspondence to the indication of interest. We illustrated this phenomenon in our Nelson et al. 2015 paper, Figure 2A. E.g. we showed that genes that had any OMIM entries for rare disease were 8x enriched among approved drugs, without considering the drug indication or genetic disease. The authors did not provide data in the supplementary materials to explore this in their data set, but I did some back-of-the-envelope calculations to explore this. From our 2023 preprint (Minikel et al.), there are ~2500 unique drug targets that progressed to the clinical since 2000. The authors report that they generated genetic support for 3,654,109 gene-trait combinations. That is on average 1,500 associated traits per drug target. This method is what leads the authors in another publication to demonstrate 2/3 of approved drugs in 2021 had genetic evidence, whereas our methods would conclude ~20% have genetic evidence. My primary concern is that the authors definition of genetic evidence is effectively any genetic evidence, mostly unrelated to the indication. There is certainly value in this definition, but the results would need to be presented, interpreted, and discussed very differently than they currently are.

Page 4, paragraph 1: There is discussion of the fraction of trials stopping with oncology indications. I was unable to explore this from the supplementary material provided. Could the indication or at least the major indication area (at very least oncology and non-oncology) be included in Stable 5? It would also help to interpret this paragraph if we were provided with the count and fraction of trials by phase, stratified by oncology and non-oncology.

Page 8: Thank you for the new supplementary figure 8, stratifying the impact of gene constraints by oncology and non-oncology. I feel strongly that these are the results that should be presented in the main section. If you look closely at the point estimates, they are generally more extreme in all trials combined, suggesting that there is a confounding/interaction between these variables under study and therapy area. You can test for this interaction effect and model them if you would like. But based on my understanding of what you have done and shown, the combined point estimates are invalid and not appropriate to present as the top-line study results.

Minor Comments

Supplementary Figure 3: should be stratified by phase, as trial duration differences by phase may be important to interpret. E.g. why does Covid19 play such a minor role in oncology? Do we need to compare to trials that didn't stop to better interpret these?

Figure 2: It would be easier to interpret and compare the results between human and mouse if the x axes were on the same scale.

Figure 3: Why the reversion to *** for p-values? Please report as per other figures.

Best regards,
Matt Nelson

Reviewer #2:

Remarks to the Author:

In my first review, I have already commented on the value and interest of this work, which I think is high. When it comes to the revisions, I appreciate the stratification by oncology and non-oncology indications. They have greatly expanded the documentation of the methodology. I also appreciate the responses to reviewer three addressing negative controls, and I agree with the authors' assessment of the results and complexities. Additional controls and checks could be considered, for example, considering how much the signal is dominated by the most popular target-indication pairs and / or highly multitarget drugs.

Reviewer #3:

Remarks to the Author:

The Reviewers have addressed my major comments. It appears that Figure 3 from the original submission is inadvertently still being used here in the revision and should be updated (Minor comments 2 and 5).

Major comments

1. Ok
2. Ok.
3. Ok.
4. Ok. I agree with the authors explanation for the exceptions of "Success" or "Insufficient data" due to low power (actually, the n (mean size of intersections) for these classes is only 1)

Minor comments

1. Ok.
2. Figure 2 now includes the p-values. Figure 3 however still has the binned representation using stars. For consistency, suggest changing to p-values. See minor comment 5 below.
3. Ok.

4. Ok.

5. Figure 3 appears to be unchanged from the original submission. I think the old Figure was inadvertently copied here for this revision.

Author Rebuttal, first revision:

Reviewers' Comments:

Reviewer #1:

Remarks to the Author:

I appreciate the additional work the authors have done to address the questions and recommendations provided by me and the other reviewers. The manuscript is substantially improved and it is much easier to follow/dissect the study. While I am still supportive of this work, I have a few additional concerns with how the study was conducted or communicated that should be considered.

Once again, we thank the reviewer for the constructive comments and suggestions and we are glad to read the updates that helped him to understand the scientific findings. We also want to apologise for the delayed response, hoping this set of changes will enrich the final manuscript. We will be happy to add any additional clarification if this is required.

Major comments

The definition of genetic support appears to be very broad, which concerns me that the associations being observed are more of a reflection that genes for terminated programs are just more likely to have genetic associations of some sort, rather than some correspondence to the indication of interest. We illustrated this phenomenon in our Nelson et al. 2015 paper, Figure 2A. E.g. we showed that genes that had any OMIM entries for rare disease were 8x enriched among approved drugs, without considering the drug indication or genetic disease. The authors did not provide data in the supplementary materials to explore this in their data set, but I did some back-of-the-envelope calculations to explore this. From our 2023 preprint (Minikel et al.), there are ~2500 unique drug targets that progressed to the clinical since 2000. The authors report that they generated genetic support for 3,654,109 gene-trait combinations. That is on average 1,500 associated traits per drug target. This method is what leads the authors in another publication to demonstrate 2/3 of approved drugs in 2021 had genetic evidence, whereas our methods would conclude ~20% have genetic evidence. My primary concern is that the authors definition of genetic evidence is effectively any genetic evidence, mostly unrelated to the indication. There is certainly value in this definition, but the results would need to be presented, interpreted, and discussed very differently than they currently are.

The reviewer's central point focuses on our definition and understanding of genetic evidence and whether this might mislead the reader in the context of these findings. We believe it's worth dissecting some of these points one by one.

One of the arguments from the reviewer is that our systematic collection of genetic evidence could be unspecific, leading to an excess of genetically-supported drug targets in a phenotype-independent manner. After performing some calculations the reviewer concluded that because we report 3,654,109 gene-trait combinations after ontology expansion, that would lead to an average of 1,500 associated traits per drug target based on his estimates that ~2500 unique drug targets progressed to clinical studies since 2000. We sincerely appreciate the effort from the reviewer, because it helped us understand the point. The first thing we need to clarify is that the 3,654,109 genetically supported target-disease pairs are not related to any clinical information. They represent any piece of evidence we were able to collate that nominates a

gene with any trait including diseases, endo-phenotypes, quantitative traits, medical procedures and several other cohort definitions. Additionally, we performed the ontology expansion which aims to ensure we maximise the overlap between the genetic information and the clinical data. Although this method might be misunderstood as a source of promiscuous mappings, on our hands, it has proven to be a useful tool to ensure parent-child subtypes of disease are properly mapped when aligning multiple sources (e.g. IBD and Crohn's Disease). When we intersect the 3,654,109 gene-disease evidence with our list of drug targets and indications, the number of clinically supported gene-disease relationships is reduced to 71,419 corresponding to 1,460 unique drug targets and 2,502 unique traits. This number denotes the majority of genetically supported gene-trait pairs that do not overlap with any pharmacological targets or disease indications. For example, the number of genes that have been knocked out in mice according to IMPC exceeds 11,000 but only a small proportion of them represent clinically studied targets. Similarly, quantitative phenotypes or medical procedures have no intersection with the clinical indications. Within the resulting 71,419 target-disease pairs, drug targets have a much more moderate number of phenotypes associated (median=4, average=12), far below the 1,500 associated traits per drug target estimated by the reviewer. To ensure our compilation of genetic evidence is not misunderstood as support for a clinically relevant hypothesis, we have clarified the origin of this number and relabeled "Target-Disease" to "Gene-Trait" associations when referring to the genetic support dataset described in Supplementary Table 4.

We want to emphasise that all our analyses have been performed using gene-disease indication pairs, so we have not aimed to reproduce the subset of the analysis from Nelson et al. 2015 in which associated genes were evaluated independently of the indication. Hopefully, the more reasonable number of associated traits per target will reassure the reviewer of the specificity of that evidence. Regarding the comment about reproducibility, we took this very seriously during the revision. Supplementary Table 9 should have all the necessary information to extract the numbers the reviewer is requesting. Because of the file size, it had to be uploaded as a different asset than the rest of the tables but it will be available to all readers. This table has been renamed from Supplementary Table 8 to Supplementary Table 9 due to changes during this revision.

Overall, the reviewer raised concerns about whether the genetic evidence used in this analysis is sufficiently specific for the given gene indication pairs. He is extending the same concern to a separate publication in which a subset of the authors of the submitted manuscript reported that two-thirds of drugs approved in 2021 had genetic backing. In contrast, the reviewer's analysis suggests that only approximately 20% of drugs are supported by genetic evidence. To bring some light to this debate, we embarked on a more detailed analysis of the causes behind the disagreement. In our manuscript - Rusina et al. 2023 (Nature Reviews Drug Discovery) - we performed a manual curation of the genetically supported drug targets throughout the last decade of FDA approvals. This is an extension of another manuscript in which we concluded $\frac{2}{3}$ of 2021 FDA-approved drugs presented genetic support (Ochoa et al. 2022 Nature Reviews Drug Discovery). There are some methodological differences between these 2 published manuscripts, the submitted manuscript and the Minikel *et al.* medrxiv preprint contributed by the reviewer. Because both NRDD pieces focus on drug approvals as opposed to

all the clinical pipeline, we manually curated the 428 drugs approved throughout the period. In each of these approved drugs, we manually confirmed that there was genetic evidence we could trust and surpass a reasonable level of certainty (e.g. common variation within a reasonable distance from the gene, reasonable pathogenicity evidence in ClinVar, etc.). Throughout the curation process, we could also find endophenotypes and quantitative traits that can act as proxies and for which genetic evidence was available (e.g. microalbuminuria for CKD). Because of the manual nature of the curation, we are quite confident about the estimates in both articles and we would welcome any feedback on any lack of confidence in the associated evidence included in the supplementary materials. Similar to the reviewer's analysis, we also obtained moderately low coverage when performing this analysis using the same high-quality data sources but performing the analysis programmatically. We believe there is a considerable gap in the interpretation of the available genetic information depending on the methodology, as we describe in the next section.

To better understand the impact of the methodological design on capturing genetically supported targets, we used as a reference the last decade of approved drugs and we evaluated the % of genetically supported therapies using different definitions of genetic evidence.

Figure 1. Genetic support for drug approvals throughout 2013-2022 based on different definitions of genetic evidence. The percentage of FDA-approved drugs is relative to the total number of approved drugs, not just those with available pharmacological targets. Different tiers of genetic evidence correspond to **T/D (exact)**: evidence available for the exact target-disease pair; **T/D (propagated)**: evidence available for the exact target and the disease of interest or any other sibling term in the disease ontology tree; **T/D + relatedD (propagated)**: evidence available for the exact target and either the disease of interest, any other sibling term in the disease ontology tree, or a manually curated phenotype relevant to the disease; **T + interactorT / D + relatedD (propagated)**: same as previous but including genetic evidence for physically interacting partners of the pharmacological target. (More methodological details in Rusina et al 2023).

As illustrated in Figure 1, how genetic evidence is defined and curated can have a big impact on the percentage of approved drugs with genetic support. In the submitted study, we are only leveraging the available genetic evidence for the exact target and the exact disease or any other

term in the disease ontology tree - *T/D (propagated)* - so we are not expecting percentages of support near the previously claimed $\frac{2}{3}$, as they would require additional curation. When revising the methodology of Minikel et al. 2013 or Trajanoska *et al.* 2023, they both mimic a setup that it's similar to "*T/D (exact)*" in which they leverage different data sources but in both cases require an exact match for the target and the disease. The other big methodological difference is that both manuscripts focused specifically on non-oncology indications. In Figure 2, we repeated the analysis as a fraction of all drugs approved for non-oncology indications in an attempt to reproduce previously reported percentages.

Figure 2. Genetic support for drugs approved for non-oncology indications throughout 2013-2022 based on different definitions of genetic evidence. The definition of genetic evidence is explained in **Figure 1**.

As reflected in Figure 2, a "*T/D (exact)*" strategy only for non-oncology indications can lead to percentages in the 20-30%, a range that is compatible with the reviewer's observations. In light of this additional analysis, we conclude different estimates are most likely derived from methodological differences. We believe a true estimate on supported approved drugs can only be accurate with a reasonable amount of manual curation both on the clinical and the genetic side. Whereas we performed curation for the 428 approved clinical studies in a separate study, it goes beyond our capacity to curate the 28,561 stopped studies. Nonetheless, we believe obtaining a precise estimate of genetic support is less crucial in this study since our primary objective is to compare the differences in the availability of evidence for various reasons leading to the discontinuation of the clinical trials. Because of the comparative nature of our analysis, we conclude the originally described methodology is precise. To minimise confusion, we have added an extra paragraph to the Supplementary Methods to clarify the nature of the genetic dataset. We are happy to clarify any additional points if any concerns would emerge.

While we think this discussion perhaps goes beyond the scope of this publication, we appreciate the community might be confused by the different numbers reported by different manuscripts. We would be happy to address this issue separately from this publication, perhaps by providing a consensus view of several authors who have worked on this topic. We are happy to start that conversation if the reviewer and/or editor are interested.

Page 4, paragraph 1: There is discussion of the fraction of trials stopping with oncology indications. I was unable to explore this from the supplementary material provided. Could the indication or at least the major indication area (at very least oncology and non-oncology) be included in Stable 5? It would also help to interpret this paragraph if we were provided with the count and fraction of trials by phase, stratified by oncology and non-oncology.

The studies belonging to oncology or non-oncology indications have been added to Supplementary Table 5 as requested. It's important to flag that our annotation of the therapeutic area is limited to these studies in which we also have a pharmacological target for the drug. To assist in the understanding of these groupings and complement Figure 1, we have incorporated a new Supplementary Table 8 with the number studies aggregated by phase and therapeutic category.

Page 8: Thank you for the new supplementary figure 8, stratifying the impact of gene constraints by oncology and non-oncology. I feel strongly that these are the results that should be presented in the main section. If you look closely at the point estimates, they are generally more extreme in all trials combined, suggesting that there is a confounding/interaction between these variables under study and therapy area. You can test for this interaction effect and model them if you would like. But based on my understanding of what you have done and shown, the combined point estimates are invalid and not appropriate to present as the top-line study results.

As per the suggestion from the reviewer, Supplementary Figure 8 is now available in the main text as Figure 3. We agree with the reviewer that this is a richer representation of the problem under study and the additional tests evaluated by “oncology” and “non-oncology” present a more rigorous understanding of the relationships between trials stopped due to safety or side effects and the respective target properties. One of the limitations of breaking down the groups by therapeutic area is the relatively low number of non-oncology studies stopped due to safety. This observation aligns with an increased tolerance for target selection risks in oncology studies as opposed to non-oncology. Despite the low numbers, we believe displaying all the tests can help inform the reader on how and why we concluded that the signals are mainly driven by oncology studies.

Minor Comments

Supplementary Figure 3: should be stratified by phase, as trial duration differences by phase may be important to interpret. E.g. why does Covid19 play such a minor role in oncology? Do we need to compare to trials that didn't stop to better interpret these?

Our intention with this supplementary material was to reflect that the predicted stopped reasons capture the expected overall complexities in the clinical setting that might differ from one therapeutic area to another. Once again, breaking down stopped reasons by therapeutic area reduces the number of observations limiting the interpretation. For this reason, we limited our analysis to the reasons displayed in Supplementary Figure 3.

Our interpretation of the results is that respiratory units were under extraordinary levels of stress during the pandemic limiting their ability to perform clinical studies for respiratory conditions. Thus, respiratory studies had a higher relative incidence of stoppage due to COVID-19 than any other therapeutic area in the analysis. On the contrary, oncology studies were rarely stopped due to COVID-19 as the main reason. The reasons behind this observation are likely to be complex but they might relate to an increased operational resilience, commercial interest and patient engagement in oncology studies.

We think Supplementary Figure 3 is a side analysis that complements the main findings. We hope this will help the reader understand there are differences in the predicted stopped reasons by therapeutic area but we will always be limited by the low numbers in some of the therapeutic areas. We would be happy to remove Supplementary Figure 3 if there is a sense it might confuse the reader.

Figure 2: It would be easier to interpret and compare the results between human and mouse if the x axes were on the same scale.

Figure 2 has been updated to include the same axis scale in both panels. Similarly, we also adjusted the axis of the newly added Figure 3 to make the plots more comparable.

Figure 3: Why the reversion to *** for p-values? Please report as per other figures.

Figure 3 has been updated to address this issue. This was our mistake during the previous submission.

Best regards,
Matt Nelson

Thanks again for the fruitful discussion.

Reviewer #2:

Remarks to the Author:

In my first review, I have already commented on the value and interest of this work, which I think is high. When it comes to the revisions, I appreciate the stratification by oncology and non-oncology indications. They have greatly expanded the documentation of the methodology. I also appreciate the responses to reviewer three addressing negative controls, and I agree with the authors' assessment of the results and complexities. Additional controls and checks could be considered, for example, considering how much the signal is dominated by the most popular target-indication pairs and / or highly multitarget drugs.

We want to use this opportunity to thank the reviewer again for the feedback provided which we believe has helped improve the quality of the publication. We will consider the reviewer's considerations on potential biases and confounders for our future research.

Reviewer #3:

Remarks to the Author:

The Reviewers have addressed my major comments. It appears that Figure 3 from the original submission is inadvertently still being used here in the revision and should be updated (Minor comments 2 and 5).

Indeed this was our mistake during the submission of the manuscript. Figure 3 has been amended.

Major comments

1. Ok

2. Ok.

3. Ok.

4. Ok. I agree with the authors explanation for the exceptions of "Success" or "Insufficient data" due to low power (actually, the n (mean size of intersections) for these classes is only 1)

Minor comments

1. Ok.

2. Figure 2 now includes the p-values. Figure 3 however still has the binned representation using stars. For consistency, suggest changing to p-values. See minor comment 5 below.

3. Ok.

4. Ok.

5. Figure 3 appears to be unchanged from the original submission. I think the old Figure was inadvertently copied here for this revision.

Thank you for pointing it out and for going again through the revised manuscript.

Decision Letter, second revision:

Our ref: NG-A61987R2

14th Mar 2024

Dear David,

Thank you for submitting your revised manuscript "Why Clinical Trials Stop: The Role of Genetics" (NG-A61987R2). It has now been seen by the original referees and their comments are below. The reviewers find that the paper has improved in revision, and therefore we'll be happy in principle to publish it in Nature Genetics, pending minor revisions to satisfy the referees' final requests and to comply with our editorial and formatting guidelines.

Sincerely,

Michael Fletcher, PhD
Senior Editor, Nature Genetics
ORCID: 0000-0003-1589-7087

Reviewer #1 (Remarks to the Author):

I appreciate the very thoughtful and thorough response to the questions I raised in my previous review, particularly regarding the definition of genetic evidence. Their clarification of the definition and the counts resolve my confusion and the additional breakdown under different definitions of evidence was very insightful.

After reviewing the response, the manuscript changes, and the additions to the supplementary information, I have no further material concerns. As previously stated, this is a significant contribution to the field and provides a tool that will hopefully be valued and reused well beyond its applications to genetic evidence.

Best regards,
Matt Nelson

Final Decision Letter:

2nd Jul 2024

Dear David,

I am delighted to say that your manuscript "Genetic factors associated with reasons for clinical trial stoppage" has been accepted for publication in an upcoming issue of Nature Genetics.

Your paper will be published online after we receive your corrections and will appear in print in the next available issue. You can find out your date of online publication by contacting the Nature Press Office (press@nature.com) after sending your e-proof corrections.

Please note that *Nature Genetics* is a Transformative Journal (TJ). Authors may publish their research with us through the traditional subscription access route or make their paper immediately open access through payment of an article-processing charge (APC). Authors will not be required to make a final decision about access to their article until it has been accepted. Find out more about Transformative

Journals

Authors may need to take specific actions to achieve compliance with funder and institutional open access mandates. If your research is supported by a funder that requires immediate open access (e.g. according to Plan S principles) then you should select the gold OA route, and we will direct you to the compliant route where possible. For authors selecting the subscription publication route, the journal's standard licensing terms will need to be accepted, including [a href="https://www.nature.com/nature-portfolio/editorial-policies/self-archiving-and-license-to-publish](https://www.nature.com/nature-portfolio/editorial-policies/self-archiving-and-license-to-publish). Those licensing terms will supersede any other terms that the author or any third party may assert apply to any version of the manuscript.

If you have not already done so, we strongly recommend that you upload the step-by-step protocols used in this manuscript to [protocols.io](https://www.protocols.io). [protocols.io](https://www.protocols.io) is an open online resource that allows researchers to share their detailed experimental know-how. All uploaded protocols are made freely available and are assigned DOIs for ease of citation. Protocols can be linked to any publications in which they are used and will be linked to from your article. You can also establish a dedicated workspace to collect all your lab Protocols. By uploading your Protocols to [protocols.io](https://www.protocols.io), you are enabling researchers to more readily reproduce or adapt the methodology you use, as well as increasing the visibility of your protocols and papers. Upload your Protocols at <https://www.protocols.io>. Further information can be found at <https://www.protocols.io/help/publish-articles>.

Sincerely,

Michael Fletcher, PhD
Senior Editor, Nature Genetics
ORCID: 0000-0003-1589-7087